# EMOS: Embodiment-aware Heterogeneous Multi-robot Operating System with LLM Agents

**Junting Chen**[*1], **Checheng Yu**[*17], **Xunzhe Zhou**[*18], **Tianqi Xu**[4], **Yao Mu**[†12],
**Mengkang Hu**[23], **Wenqi Shao**[3], **Yikai Wang**[‡6], **Guohao Li**[5], **Lin Shao**[‡1]

[1]National University of Singapore, [2]The University of Hong Kong, [3]Shanghai AI Laboratory,
[4]KAUST, [5]University of Oxford, [6]Tsinghua University, [7]Nanjing University, [8]Fudan University

## ABSTRACT

Heterogeneous multi-robot systems (HMRS) have emerged as a powerful approach for tackling complex tasks that single robots cannot manage alone. Current large-language-model-based multi-agent systems (LLM-based MAS) have shown success in areas like software development and operating systems, but applying these systems to robot control presents unique challenges. In particular, the capabilities of each agent in a multi-robot system are inherently tied to the physical composition of the robots, rather than predefined roles. To address this issue, we introduce a novel multi-agent framework designed to enable effective collaboration among heterogeneous robots with varying embodiments and capabilities, along with a new benchmark named Habitat-MAS. One of our key designs is *Robot Resume*: Instead of adopting human-designed role play, we propose a self-prompted approach, where agents comprehend robot URDF files and call robot kinematics tools to generate descriptions of their physics capabilities to guide their behavior in task planning and action execution. The Habitat-MAS benchmark is designed to assess how a multi-agent framework handles tasks that require embodiment-aware reasoning, which includes 1) manipulation, 2) perception, 3) navigation, and 4) comprehensive multi-floor object rearrangement. The experimental results indicate that the robot's resume and the hierarchical design of our multi-agent system are essential for the effective operation of the heterogeneous multi-robot system within this problem context. The project website is: `https://emos-project.github.io/`

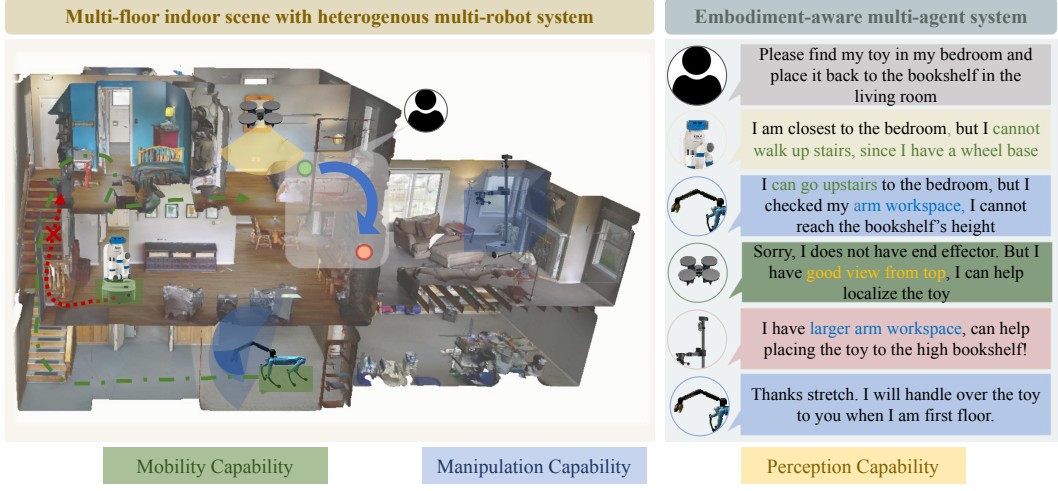

Figure 1: **Embodiment-aware LLM-based MAS.** This figure depicts how an LLM-based MAS operate a HMRS composed of dones, legged robots and wheeled robots with robotic arms, in a multi-floor house. When given a household task, the LLM-based MAS needs to undertand their respective robots' hardware specifications for task planning and assignment. The authors refer this capability as "embodiment-aware reasoning" in this work.

[*]These authors contributed equally to this work.

[†]Yao Mu participated in this work during his internship at the National University of Singapore.

[‡]Corresponding author: Yikai Wang(wangyk17@mails.tsinghua.edu.cn), Lin Shao (linshao@nus.edu.sg)

## 1 INTRODUCTION

The complex nature of real-world environments and specialized robot hardware makes it difficult for a single robot to perform complex tasks efficiently. As a result, the Heterogeneous Multi-Robot System (HMRS) has emerged, enabling multiple robots designed for diverse purposes and with complementary physics capabilities to cooperate and execute complex missions through task decomposition, coalition formation, and coordinated task allocation. Designed for real-world deployments, existing HMRSs are highly dependent on some assumptions and human-crafted protocols based on human prior knowledge (Rizk et al., 2019). This limits the generalization of HMRS and the ability to handle complex tasks. In the survey, Rizk et al. (2019) classified the automation of HMRS into four levels: 1) Level 1, task execution. 2) Level 2, task execution plus task allocation or coalition formation, but not both. 3) Level 3, the automation of all above but not instruction to decomposed sub-tasks. 4) Level 4, fully automated entire system. To the best of our knowledge, no system has achieved an automation level of 4.

Meanwhile, we have recently witnessed how large language models (LLM) multi-agent systems (MAS) operate complex systems like Operating Systems (Mei et al., 2024) or finish complex tasks like software development (Hong et al., 2023), by leveraging the common sense reasoning capabilities and code generation capabilities to generally control diverse applications. Similarly in embodied AI tasks, Mandi et al. (2024) proposed using LLM-based MAS chat to control the duel arm system. Zhang et al. (2023) introduced a human-robot collaboration system through the LLM multi-agent. These works focused on certain aspects of MRS automation problem or focused MRS with specific hardware configuration. Our observation is that one missing key component toward the level-4 full automation is *embodiment-aware reasoning*. It refers to the agent's ability to understand its physical embodiment and thus the hardware-dependent capabilities. Based on this capability, the LLM multi-agent can further decompose the tasks, assign the tasks, and finally execute the tasks in the real time, i.e. the level-4 automated HMRS.

In this work, we propose EMOS, a general LLM-based multi-agent framework to operate cooperative HMRS in indoor household environments. Our insight is that, instead of teamwork through role assignment as in recent LLM-based MAS (Hong et al., 2023; Li et al., 2024; Wu et al., 2023a), the LLM-based MAS tailored for heterogeneous robots should actively check their physics information and tasks they can complete without fixed roles. Thus, we introduce a bottom-up robot capability generation approach that constructs a "robot resume" for each robot, capturing its unique skills and constraints. The resumes, along with a scene description and task description, form the full context for the LLM-based MAS to perform task planning, task assignment, and action execution in a cascaded manner. To study how LLM-based MAS could potentially enable the full automation of collaborative heterogeneous multi-robot systems, we present Habitat-MAS, which is a benchmark with annotated episodic data and an accompanying simulated environment with textual description of the environment as the interface for the agents. In the benchmark, we provide a diverse collection of robots including drones, wheeled robots with arms or elevatable grippers on a rack, and legged robots with arms, and also diverse environment including multi-floor large houses and multi-room flats. The benchmark presents four tasks, each designed to evaluate multi-agent systems in terms of their understanding of robot physical capabilities including perception, navigation, and manipulation. Episodes are processed such that only robots possessing specific physical abilities can successfully complete certain subtasks in an episode. Through extensive experiments, we illustrate the importance of robot resumes in embodiment-aware reasoning and how different components in EMOS affect HMRS performance in our benchmark.

To summarize, the key contributions of this paper are:

- *We present EMOS, a novel LLM-based MAS framework that first conducts embodiment-aware reasoning with self-generated robot resume, rather than human-assigned role playing, to operate a collaborative HMRS.*
- *We present Habitat-MAS, a new benchmark to study how LLM-based MAS can coordinate collaborative HMRS. To the best of our knowledge, this is the first simulated benchmark for this problem with extensive robot types and scenes. It is also highlithed as the first benchmark to evaluate the agent's understanding of its physics embodiment, with test dataset tailored for this purpose.*
- *Experimental results on Habitat-MAS demonstrate the effectiveness of robot resume in EMOS, highlighting the significance of embodiment awareness for collaborative HMRS.*

## 2 RELATED WORK

**LLM-Based Multi-Agent System.** The integration of LLMs into MAS is a relatively new yet rapidly growing area of research. This integration leverages the language understanding and generation capabilities of LLMs to enhance communication, coordination, and decision-making within MAS. Wu et al. (2023a); Hong et al. (2023); Li et al. (2024) focus on the communication issues in LLM-based Multi-Agent Systems. Xu et al. (2024) proposes Crab, a cross-environment benchmark framework for evaluating Multimodal Language Models (MLMs) in different GUIs like mobile phones and desktop computers. For robotic intelligence, Zhang et al. (2023) investigates how two agents can use communication to better collaborate and complete tasks in a multi-room scenario. Mandi et al. proposed RoCo, which is a multi-agent system for multi-arm collaboration. They try to leverage the 3D spatial reasoning capabilities to help multi-arm low-level trajectory planning. In comparison, we focus a more general mutli-agent scenario with drones, legged robots, wheeled robots with arms, with a multi-agent system required to understand general capabilities including navigation, manipulation and perception based on physics design.

**Heterogeneous Multi-Agent Learning.** Heterogeneous multi-agent systems involve agents with varying capabilities or functional roles working collaboratively toward shared goals. This field has gained significant attention due to its practical relevance in real-world applications requiring diverse agent teams. Recent works have introduced innovative frameworks for learning and coordination in heterogeneous teams. For example, Seraj et al. (2022) proposes a method for learning communication protocols tailored to each robot's role and capabilities, optimizing team performance in dynamic environments. Similarly, Bettini et al. (2023) develops reinforcement learning algorithms specifically designed for heterogeneous teams, enabling effective inter-agent coordination despite differences in robot traits. Task allocation is another critical aspect. Ravichandar et al. (2020) provides a scalable optimization framework for balancing workload and resource utilization across large heterogeneous teams. In addition, Seraj et al. (2024b) combines human demonstrations with machine learning to train diverse robot teams efficiently. Seraj et al. (2024a) introduces a novel policy network architecture that integrates individual robot policies into a composite framework for effective decision-making. Our work distinguishes these works in the way that we handle the behavior of these heterogeneous agents by transfering the prior knowledge in the pre-trained large-language models without extra training.

**Multi-Robot System.** Early works by Arai et al. (2002) and Ota (2006) laid the research foundation for multi-robot systems by providing a comprehensive overview of the progress and key challenges in MRS in around 2000, including MRS architectural design, distributed mapping, and navigation coordination, etc. Hamann & Wörn (2008) proposed a model framework with an explicit space representation for swarm robotic algorithm design, deriving an abstract swarm motion model from a single robot description and validating it against simulation results, while also discussing the challenges and related work in this area. Rizk et al. (2019) specifically reviewd the challenges in cooperative heterogeneous MRS, decomposing the MRS workflow to task decomposition, coalition formation, task allocation, perception and MRS planning and control. In this work, we also follow the established concept definitions and the principles of system design in this survey. Roldán et al. (2016) built a HMRS composed of aerial vehicles (drones) and ground vehicles to collaborate to monitor environmental variables of greenhouses. Kiener & Von Stryk (2010) designs a system composed of wheeled robot and humanoid robot to collaborate in a "robot soccer" scenario. The authors carefully decompose the complex task into subtasks based on the robots' capabilities, followed by human-crafted task allocation and planning algorithms. Yang & Parasuraman (2020) proposed Self-Adaptive Swarm System (SASS), a hierarchical needs-based framework for cooperative multi-robot systems, inspired by Maslow's hierarchy of human needs, combining multi-robot capabilities with a distributed negotiation-agreement mechanism that prioritizes robots' needs according to Maslow's human needs principle.

**Task Planning With Large Language Models.** Large language models(LLMs) trained on massive corpora are generally considered to have acquired common sense knowledge for task planning (Vemprala et al., 2023; Yao et al., 2022; Zhao et al., 2023). Thanks to recent advancements, directly generating plans with LLMs has become an active research area in recent years (Logeswaran et al., 2022; Wu et al., 2023b; Lin et al., 2023). When using LLMs for task planning, some approaches directly generate the entire plan in an open-loop manner, that is, without executing it in the environment (Huang et al., 2022a; Mu et al., 2023; Singh et al., 2022). An alternative line of research

investigates closed-loop task planning, which offers greater flexibility for error correction, human interaction, and grounding the plan in the actual environmental state (Ahn et al., 2022; Guo et al., 2023; Huang et al., 2023; Hu et al., 2023; Huang et al., 2022b; Song et al., 2023; Hu et al., 2024). This paper explores closed-loop task planning, where real-time environmental changes are integrated, and a central large language model processes these real-time changes and adapts plans accordingly.

## 3 EMOS FRAMEWORK

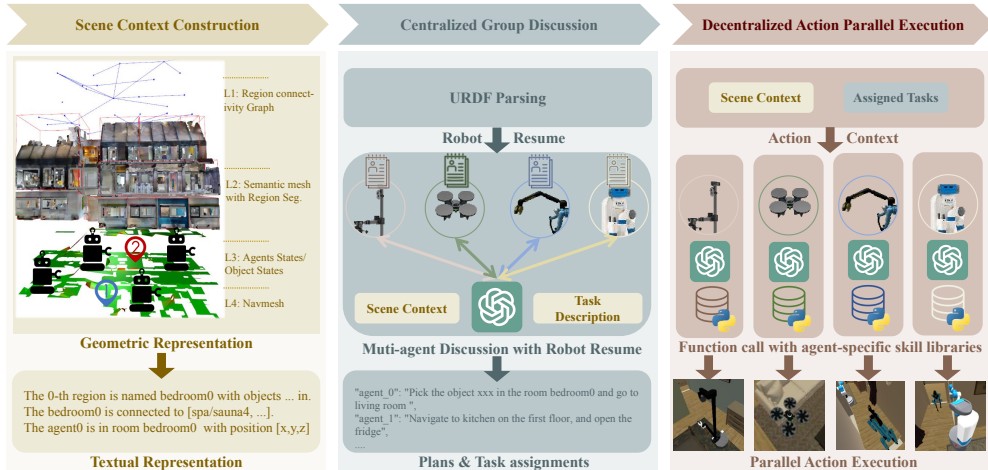

Figure 2: **EMOS Framework.** This figure illustrates how EMOS operates an HMRS on the Habitat-MAS platform. There are three stages: 1) Scene Context Construction involves generating scene descriptions in a bottom-up approach, relying on an ideal semantic SLAM system. 2) In Centralized Group Discussion, agents perform embodiment-aware reasoning for task planning and assignment 3) In Decentralized Action Parallel Execution, agents execute actions parallely with initial context and agent history. Precisely speaking, EMOS only includes stages 2 and 3, while stage 1 is integrated inside the Habitat-MAS platform. We include it in this diagram for completeness and clarity.

The multi-agent system introduced in this paper focuses on developing an embodiment-aware framework for heterogeneous multirobot collaboration. Traditional multi-robot systems often face challenges related to coordinated motion planning, especially in complex environments involving diverse robotic platforms such as UAVs, mobile robots, legged robots, etc. The team formation and cooperative protocol are designed by robot experts, rather than automated by robots themselves. This system aims to address these challenges by enabling agents to understand robots with different physical capabilities and operational constraints. In the rest of this section, we organize the introduction to EMOS as follows: In 3.1, we will first present the overview of this system. Then in 3.2, we briefly introduce how the textual scene context is constructed from an ideal scene reconstruction. In 3.3, we will elaborate on the composition and generation pipeline of robot resumes. Finally in 3.4, we will show how EMOS performs task planning, assignment and action execution in a hierarchical fashion.

### 3.1 FRAMEWORK OVERVIEW

For clarification, we first define the mathematical form of the problem solved by the multi-agent system. Assume there is a multi-robot system involving $N$ different robots and there is an LLM agent $i$ attached to each of the robots $i$, $i \in \{1, 2, \ldots, N\}$. All agents operate in a shared environment with state space $\mathcal{S}$, and each agent $i$ has an observation space $\mathcal{O}_i$ and an action space $\mathcal{A}_i$. The multi-agent system is designed to collaboratively achieve a given task $T \in \mathcal{T}$, such as exploring an unknown environment. The system serves as a set of task-conditioned policies $\{\pi_i : \mathcal{O}_i \times \mathcal{T} \to \mathcal{A}_i\}_{i=1}^N$, where $\mathcal{T}$ represents the textual space of the task description. However, rather than an end-to-end policy network as it might hint, our proposed multi-agent system adopts a discussion-like, hierarchical framework, which has been proven effective in many other multi-agent scenarios. As Figure 2 demonstrates, the multi-agent system involves three cascading stages: 1) scene context construction;

2) centralized group discussion; and 3) decentralized action parallel execution. Since the focus of this work is embodiment-aware reasoning in task planning, we assume the multi-robot system is equipped with a perfect multi-agent SLAM system and provide the perfect geometric representation as an observation to the multi-agent system at the initial state. The geometric representation will be further processed into textual representation as the scene context for multi-agent discussion. With robot resume processed from robot URDF, the multi-agent system performs a group discussion to decompose the task and assign subtasks to corresponding agents based on their physics limitations.

## 3.2 SCENE CONTEXT CONSTRUCTION

For the deployability of the LLM multi-agent system onto the real multi-robot system, we propose a bottom-up pipeline to construct the textual scene context from the geometric representation of the environment that can be reconstructed from a normal robot perception pipeline. Following the environment representation reconstruction framework in Hydra (Hughes et al., 2022), the geometric representation is composed of four layers: 1) *L1 Region connectivity Graph* is a graph data structure, with nodes representing distinct regions in the environment and edges representing the navigational connectivity between these regions. The regions here refer to rooms and functional areas such as corridors and stairs, following the conventions in datasets Chang et al. (2017). 2) *L2 Semantic Mesh* is the direct output of a SLAM system. 3) *L3 Agent States and Object States* track the useful dynamic information in the scene for the robot-environment interaction. 4) *L4 Navmesh* is a triangle mesh for trajectory planning on its surface, commonly used in the game industry and rough terrain navigation. Although we built L1, L2, and L3 with ground truth semantic mesh and robot odometry, these layers are instantly available in Hydra-multi (Chang et al., 2023) when running on a real multi-robot system. For L4, we build the navmesh with Recast Navigation (Mononen, 2009).

To construct the textual representation of the scene, L1 and L3 are transformed into textual descriptions. In contrast, L2 and L4 are used in detailed point-to-point trajectory planning and low-level robot control. Given *L1 Region connectivity graph* $G = (V, E)$, where:

- $V$ is the set of vertices that represent distinct regions in the environment. Each node $v_i \in V$ corresponds to a specific region. Each region contains maintains the agents and objects within it.

- $E$ is the set of edges that represents navigational paths between regions. An edge $e_{ij} \in E$ exists between two nodes $v_i$ and $v_j$ if there is a direct navigable path between the region $i$ and region $j$.

The textual representation of the environment is constructed by iterating over all region nodes in the graph and checking their containing objects or robots.

## 3.3 ROBOT RESUME

**LLM-Prompted and Kinematic-Based Robot Resume Generation** A robot resume is a JSON file that contains the key hardware-specific capabilities for embodiment-aware reasoning, including 1) mobility capability, 2) perception capability and 3) manipulation capability. Each capability in a robot resume encompses two parts: 1) a comprehensive summary of the robot's capabilities in natural language and 2) a numerical representation of those capabilities. As suggested by Figure 3, a hybrid approach combining LLM summarization and forward kinematics is used to generate the robot resume from the robot URDF.

For the LLM summarization process, we first pre-process the URDF file to the *urdf tree skeleton*. This skeleton tree is a text representation of the robot's skeleton, with links as nodes and joints as edges. This step is to reduce the length of the robot's URDF, especially to remove the tags that hardly help in this step, including $< intertial >< visual >< collision >$ and etc. For complex robot URDF files with thousands of lines of code, the extreme long context could dramatically undermine the quality of the robot summary from the LLM.

For the numerical representations, we provide the forward kinematics API to load the robot URDF file so that an articulated robot can check the geometric information of the sensors and end effectors. For example, as depicted in Figure 3, the arm workspace is represented as a hulling of all sampled end effector positions in the 3D world. This numerical information is used when the multi-agent system wants to check exactly which robot in the team can interact with a certain object with its

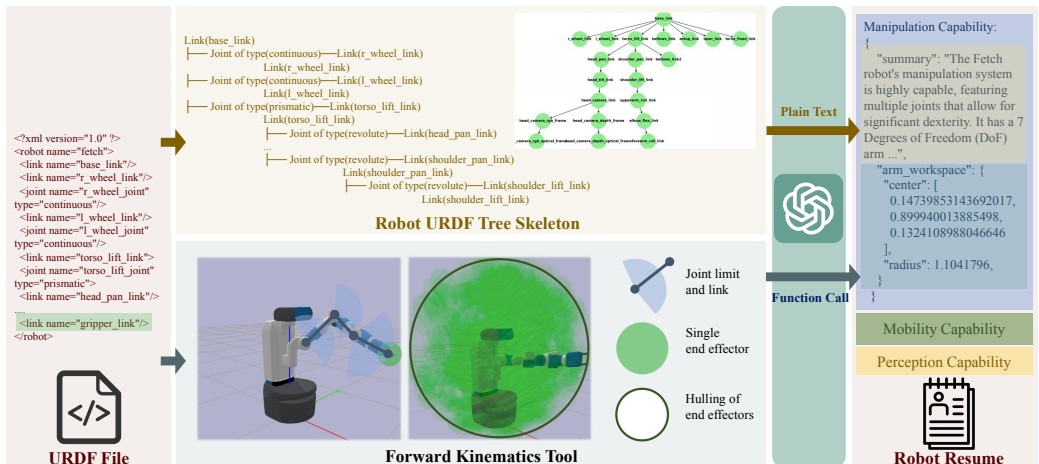

Figure 3: **Robot Resume Generation.** This figure illustrates how an LLM agent is prompted to generate a robot resume from the robot's URDF file by combining two approaches. On one hand, the LLM agent reads the skeleton of the URDF to summarize a textual description of the general capability. On the other hand, the LLM agent calls forward kinematics tool functions to generate numerical details.

positions in the 3D space. We also generate the mobility capability and perception capability in a similar way. For detailed capability definition, please refer to Appendix A.1.

## 3.4 HIERARCHICAL TASK PLANNING, ASSIGNMENT AND ACTION

To adapt the LLM-based MAS for real-time HMRS operation, the multi-agent action policy needs to be asynchronous, due to the potential asynchrony in multi-robot action execution. For this purpose, we design a hierarchical pipeline to perform task planning, assignment, and action execution for our LLM-based MAS. Specifically, there are two stages: 1) The first stage of Centralized Group Discussion runs in a synchronized fashion, in which all agents wait for messages from other agents, and the discussion history is seen by all agents. 2) While in the second stage of Decentralized Action Execution, each agent generates an action, waits for its execution it in the world, and generates a new action so on so forth. Each robot is associated with a robot-dedicated agent with full access to its robot resume to assist in decision-making and action execution. The pseudocode in Algorithm 1 provides a comprehensive overview of the hierarchical task planning, assignment, and then action execution within the EMOS framework. For detailed explanation for Algorithm 1 and MAS design, please refer to Appendix A.2

## 4 HABITAT-MAS BENCHMARK

Habitat-MAS 4 is a benchmark designed to evaluate LLM multi-agent systems (MAS) deployed in collaborative heterogeneous multi-robot systems (MRS) in multi-floor household scenarios. The LLM multi-agent system needs to do task planning, task assignment, and action execution with the comprehensive understanding of the robot physics capabilities and task-relevant environmental information to succeed in the tasks. The setting reflects real-world robotic challenges, where agents with varying embodiments, such as wheeled, legged, and aerial robots, must cooperate to accomplish complex tasks that require different physics capabilities.

### 4.1 BENCHMARK OVERVIEW

The Habitat-MAS benchmark is based on Habitat ((Puig et al., 2023)), a highly configurable simulation platform for embodied AI challenges that extensively supports the integration of various indoor environment datasets. For diversity, we choose to build the Habitat-MAS benchmark on multi-floor real-scan scenes in Matterport3D (Chang et al. (2017)) and single-floor synthesized scenes in HSSD

---

**Algorithm 1:** Hierarchical Task Planning, Assignment and Action in a Multi-Agent System

---

**Input:** Set of robots $R = \{r_1, r_2, \ldots, r_n\}$, Robot resumes $\{resume_1, resume_2, \ldots, resume_n\}$, Task $T$
**Output:** Task completion status

1   **Stage 1: Centralized Group Discussion**
2   $subtask_i \leftarrow$ CentralPlanner($T, resume_i$) ;            // Central LLM assigns a task to each robot
3   **foreach** *robot* $r_i \in R$ **do**
4      $feedback_i \leftarrow$ Reflection($subtask_i, resume_i$) ;    // Robot-dedicated agent reflects the subtask
       feasibility and gives feedback
5      **if** $feedback_i$ *is invalid* **then**
6         Reassign $subtask_i$ ;                // Central planner adjusts based on feedback

7   **Stage 2: Decentralized Action Execution**
8   **foreach** *robot* $r_i \in R$ **do**
9      $history_i \leftarrow []$
10     **while** *Not* TaskFinished($r_i$) **do**
11       $action_i \leftarrow$ FunctionCall($r_i, subtask_i, history_i$) ;     // Select current action by function
        calling
12       $response_i \leftarrow$ ExecuteAction($r_i, action_i$) ;        // Execute the action in simulation
13       $history_i \leftarrow [history_i, action_i, response_i]$
14       **if** TaskFinished($r_i$) **then**
15         WaitState($r_i$) ;               // Transition to wait state after task completion

16   **if** *All robots are in* WaitState **then**
17      **return** *Done*

---

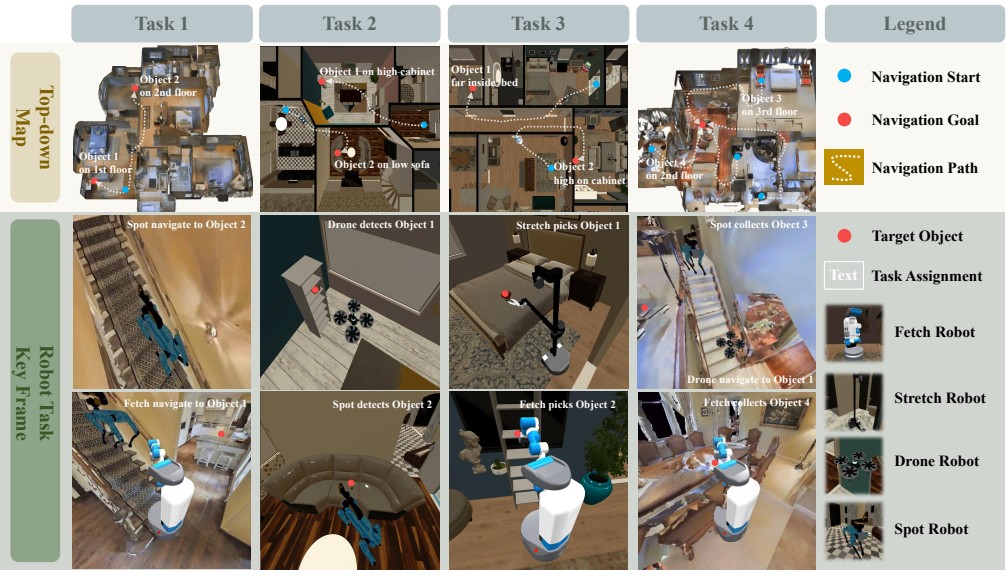

Figure 4: **Habitat-MAS Benchmark.** The figure demonstrates the four tasks (columns) from two indoor scene datasets, HSSD (Khanna* et al., 2023) and Matterport 3D (Chang et al., 2017). The upper row demonstrates the top-down maps of the environment and the successful navigation paths of the tasks. The middle and bottom rows depict the key frames of the tasks in the third-person view when robots perceive or manipulate the target objects.

(Khanna* et al. (2023)). In our full dataset, we cover 27 scenes in Matterport3D and 34 scenes in HSSD. There are four base robot types in the benchmark: 1) **Fetch** has a wheeled base and a 7-DOF arm of revolute joints. 2) **Stretch** has a wheeled base and a telescoping arm of prismatic joints. 3) **Drone** is in fact a DJI M100 with an RGBD sensor for its model credit. Since we care more about high-level discrepancies across different types of robot in the multi-robot systems, we neglect more specifications of the M100 compared to other drones. 4) **Spot** has a legged base with 7-DOF arm of the revolute joints. All end effectors are two-finger grippers.

The benchmark provides well-defined APIs for both task planning and robot control. For task planning and assignment, agents have access to tools for robot resume access construction and a

Python code interpreter to execute the code during the first stage. Besides, for the robot control, robot agents have access to APIs that link to low-level robot skills, including fundamental actions such as `navigate_to`, `move_arm_to`, `pick`, and `place`, etc. Temporarily, these low-level robot skills are implemented with a classic robot trajectory planner and inverse kinematics solver. There are certain limitations in the skill libraries, such as the absence of explicit gripper control, which are addressed by contact-based grasping. As long as the robot gripper contacts an object, we snap the object to the gripper. One reason for this design is that, if we enable physics simulation for object-gripper contact simulation, it will introduce more irrelevant failures depending on the parameter tuning of physics simulation integration. Thus, we disable the integrated Pybullet (Coumans & Bai (2016–2021)) physics simulation in the benchmark. However, for tasks that require more sophisticated low-level robot control, the benchmark can be easily extended with re-enabling the physics simulation and robot learning-based policies. For more detailed explaination about simulation and robot low-level control, please refer to Appendix B.4.

The benchmarking data are stored in task episodes. Each task episode is a snapshot of the start state and goal state of a scene and an MRS with a specific task. The MRS is anticipated to roll out a policy episode to complete the task in the environment and achieve the goal state. We note a task episode as $\mathcal{E} = (L, \mathbf{P}^0, T, G)$, composed of the starting-state scene layout $L$, initial world frame robot states $\mathbf{P}^0 = \{\mathbf{P}_i^0\}_{1 \leq i \leq N}$, task description $T$, and world goal state $G$.

## 4.2 TASK OVERVIEW

There are four tasks carfully designed in the Habitat-MAS benchmark. Tasks 1, 2, and 3 aim to evaluate if agents are able to understand the three aforementioned robot capabilities respectively. Specifically, 1) **Task 1** is deigned as a cross-floor object navigation task including two robots (wheeled and legged) navigating in a multi-floor scene, aiming to evaluate agent's ability to understand robot's mobility; 2) **Task 2**, named cooperative perception for manipulation, represents a common scenario in multi-robot collaboration, where robot perception assist manipulation. This task is set up to test the ability of MAS to reason about robot sensor type or view point in other word; 3) **Task 3** is a classic household rearrangement task including two robots with different manipulation capabilities collaborate to manipulate object placed on specific receptacles, this task can cleverly test MAS's ablility to understand robot arm's workspace; 4) **Task 4** is a multi-floor multi-agent and multi-object rearrangement task that requires the LLM-based multi-agent systems to comprehend all information and capabilities properly to collaborate. It is important to note that, during the creation of the benchmark dataset, we carefully filter the task episodes so that each robot in the scene can only complete a subset of the subgoals. In other words, the multi-agent system must comprehend robots' physical capabilities to forge a feasible plan. For detailed task description, refer to Appendix B.1.

## 4.3 EVALUATION CRITERIA

The performance of multi-agent collaboration in Habitat-MAS is evaluated using several key metrics: 1) **Success Rate.** Based on the task design we introduced in the last section, we define a series of intermediate subgoals in PDDL language for each task to evaluate the task result. This metric evaluates the proportion of episodes in which an MRS successfully completes all sub-goals, which directly reflects the overall planning and coordination capabilities of MAS. 2) **Sub-goal Success Rate.** This metric calculates the percentage of sub-goals achieved by the MRS. Due to the limit on pages, please refer to Appendix B.2 for more details about the sub-goal definition and implementation. 3) **Token Usage.** The used tokens are a key metric to evaluate the efficiency of LLM-based MAS. The effectiveness of agent communication and action planning is measured by the number of tokens used during discussions. This reflects how efficiently agents coordinate and strategize to complete tasks. 4) **Simulation Step.** We also evaluate the number of simulation steps consumed by the MAS to complete each task. Drones typically move the fastest, followed by wheeled robots, with legged robots being the slowest. This metric evaluates the LLM-based MAS' ability to assign tasks for high MRS efficiency. For instance, in an extreme scenario, one robot handles all the subtasks, leaving the rest of the robots without any assignments. This situation results in low efficiency for the HMRS and causes an abnormally large number of simulation steps.

Table 1: **Experimental results of EMOS and ablated methods on Habitat-MAS benchmark.**

| Method | Succ. Rate ↑ | Sub-goal Succ. Rate ↑ | Token Usage ↓ | Simulation Step ↓ |
|---|---|---|---|---|
| EMOS (Ours) | **37.82%** | **81.26%** | 80783 | 2358 |
| w/o. Numerical | 23.56% | 71.04% | 53201 | 2983 |
| w/o. Robot resume | 15.63% | 65.27% | 64600 | 3125 |
| w/o. Discussion | 15.23% | 72.45% | **36377** | **2332** |

## 4.4 EXPERIMENTS WITH EMOS

In this section, we present the experimental result of our EMOS system on our Habitat-MAS benchmark, along with ablation studies to explain the impact of different building blocks. Our benchmark offers a large-scale dataset with episodes in more than 70 distinct scenes. However, due to budget constraints, all ablation studies were conducted on a subset of 519 episodes. We use the GPT-4o (OpenAI, 2024) API of the May 2024 version in this experiment. For more details on how episodes are generated and the full set of those episodes, please refer to Appendix B.3.

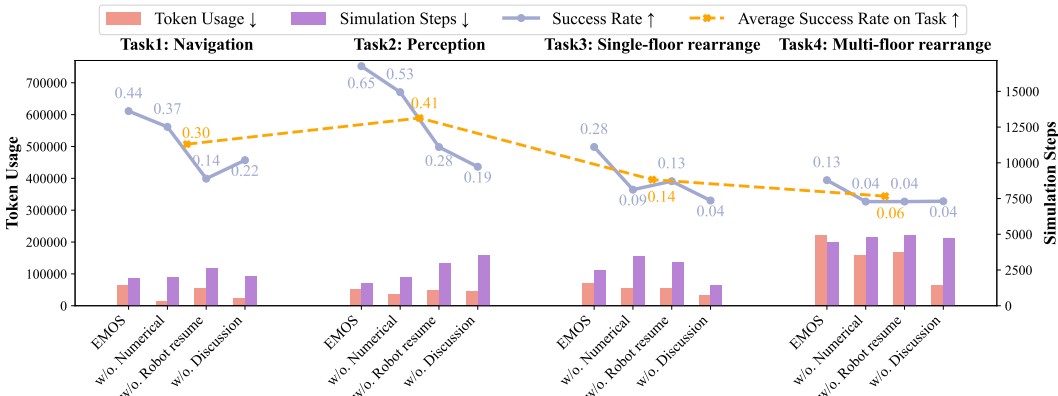

Figure 5: **Experimental Results of EMOS and Ablated Methods by Tasks.** This figure illustrates the performance of EMOS and ablated methods on the four tasks in the Habitat-MAS benchmark. The four tasks introduced in 4.2 are placed in four columns. For each task, we plot its task success rate with a blue line in the upper row, and a histogram of token usage and simulation steps in the lower row, for different ablation settings. In terms of success rate, the EMOS framework achieves a clear margin over the other ablation settings, especially the setting without robot resume. The dashed line shows the difficulty discrepancy across all four tasks. Each data point on the dashed line represents the success rate averaged over all ablation settings in this task.

The ablated methods in the experiment are as follows:

1) **EMOS:** This the multi-agent system we introduced in this work. It consists of 1) robot resume generation module, 2) centralized task planning with group discussion module, and 3) distributed action execution through function call.

2) **w/o. Numerical capability description (w/o. Numerical):** In this setting, there will be no numerical descriptions by calling the forward kinematics function tools in the robot resume. The robot agent cannot generate code to check the task geometrically, but it still has access to the robot's summarization from URDF by itself.

3) **w/o. Robot resume, with role description (w/o. Robot resume):** This ablation setting aims to provide a setting similar to role-playing multi-agent systems like Camel Li et al. (2024), MetaGPT Hong et al. (2023), etc. Robot agents do not have access to the URDF files. Instead, each robot agent possesses a role description authored by humans, which outlines their characteristics in the multi-robot system.

4) **w/o. Group discussion (w/o. Discussion)** This is a relatively dummy baseline. All robot agents receive the raw task description and scene description, and directly generate actions.

Figure 5 illustrates the primary experimental results of the methods on four tasks in Habitat-MAS benchmark respectively. The blue lines of success rate clearly demonstrate the declining trend in performance as more key modules are removed from EMOS. The pink and purple bars indicate the token usage and simulation steps for each method. While there is no unified pattern for all histograms in all tasks, we can observe the surge in both token usage and simulation steps for task 4. This is expected since task 4 involves more robots and target objects compared to task 1-3 and larger scenes compared to task 2-3. Accordingly, the dashed line of average success rate on tasks indicates the discrepancy in difficulty across all tasks. In particular, all methods demand significantly more tokens and simulation steps on the most challenging task 4.

We present the numerical results of the ablation study in Table 1. Firstly, by comparing the EMOS with EMOS w/o. numerical description, we observe that LLM agents can still perform relatively well in simple tasks like navigation and perception, as in Figure 5. We infer this is because LLM agents can still understand robots' mobility from the tree structure of URDF, recognizing the node names like wheel, leg, etc. Additionally, LLM agents can infer from their common sense that Drone is an aerial robot with a relatively broad view without any explicit information about Drone's camera height in the input URDF. However, the success rate drops significantly in more complex tasks like single-floor rearrangement ($28.35\% \rightarrow 9.20\%$) and multi-floor rearrangement ($13.46\% \rightarrow 3.85\%$). This emphasizes that mere textual descriptions are inadequate for robotic tasks requiring precise manipulation. In this case, invoking mathematical functions to process and reflect numerically helps. Secondly, in the setting w/o. robot resume, by further removing the textual summary extracted from URDF, the success rate in navigation ($37.37\% \rightarrow 14.14\%$) and perception ($52.94\% \rightarrow 28.32\%$) tasks both decrease dramatically in the experiments on MAS without robot resume. This result proves that LLM agents can indeed be aware of robot embodiment capabilities through commonsense reasoning, rather than human-assigned role-playing. By comparing the average task success rate of this ablated setting with our intact EMOS system ($37.82\% \rightarrow 15.63\%$), as in Table 1, it confirms the superiority of our methods utilizing both numerical reasoning with tools and textual reasoning with common sense. Thirdly, compared to EMOS, the setting w/o. group discussion performs the worst on success rate ($37.82\% \rightarrow 15.23\%$) as demonstrated in Table 1. In this setting, each robot agent executes tasks directly according to task description, without leader assignments or self-reflection on their embodiedment limitation using mathematical tools. Although this setting significantly reduces token usage by a large margin compared to the other three methods, it also dramatically reduces the success rate. In addition, when inspecting the column of simulation steps in Table 1, EMOS can complete the tasks in the second least steps on average, while EMOS w/o. robot resume struggles in planning, consuming the most steps, and EMOS w/o. group discussion ends the episodes with the least steps due to failure. For special case study, please refer to Appendix C.1. For more detailed discussions on experiments, see Appendix C.

## 5   CONCLUSION

In summary, this paper introduces the Embodiment-Aware Heterogeneous Multi-Robot Operating System (EMOS), an LLM multi-agent system designed to operate multi-robot systems in complex household environment. The key challenges addressed in this system are embodiment-aware reasoning and spatial reasoning in household tasks in the 3D world. The proposed framework integrates a novel "robot resume" feature that dynamically captures the physical capabilities of heterogeneous robots and uses a hierarchical, decentralized approach for task planning and execution. The system is validated through the Habitat-MAS benchmark, which includes a variety of tasks requiring robots to collaborate across different mobility, perception, and manipulation capabilities. The experimental results demonstrate the significance of embodiment-awareness and spatial reasoning in heterogeneous multi-robot systems. The ablation studies specifically highlight the importance of using numerical information for precise spatial reasoning, and group discussion modules to decompose the complex tasks in improving task success rates.

Future work could focus on system-level issues, like improving the system's scalability in multi-agent communication protocol to even more diverse robot types and a much larger number of robots (e.g., swarm system), and expanding the framework's adaptability to more dynamic, real-world settings in which the system needs to handle external disturbance or subjects with unknown intention.

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

# A   APPENDIX: ADDITIONAL DETAILS FOR EMOS

## A.1   CAPABILITIES FOR EMBODIMENT-AWARE REASONING

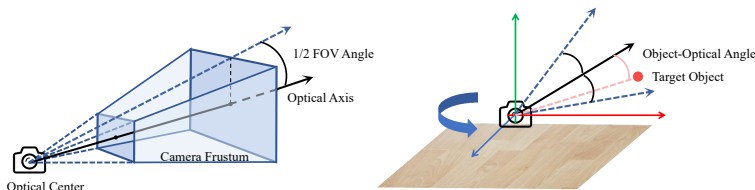

Figure 6: **Perception capability represented by camera frustum.** The left figure depicts the camera frustum of a classic perspective camera model. The right figure intuitvely demonstrates how the LLM agent reasons how possible a target object can be perceived by a camera at a certain pose.

As mentioned in the previous section, we categorize the hardware-specific capabilities into three dimensions. Our considerations are as follows:

- **Mobility.** Robots exhibit different types of mobility capabilities. For example, aerial robots like drones can move in non-occupied 3D space, legged robots like Spot can move across floors and low obstacles, and wheeled robots like Fetch and Stretch can only move on the flat ground. Thus, for a general heterogeneous multi-robot system, the deployed LLM agents should be aware of the robot's mobility capabilities when making navigation decisions. The benchmark will present a comprehensive scene description about regions and their interconnectivity to help language models deduce potential navigation pathways for navigation decision making.

- **Perception.** Robots are equipped with sensors (e.g., RGBD cameras) to perceive the environment. The perception capabilities include sensor types and camera projection models. Specifically, as shown in Figure 6, we use a simplified frustum model including the optical axis and camera Field-Of-View (FOV) angle defined in Equation 1, where $x$ represents the distance from the camera's central axis and $f$ represents the focal length. The agent needs to be aware of the robot's perceptual space and check if objects to perceive can potentially fall within the camera frustum. For example, due to jaw camera height and angle limitation, the Spot lacks the capability to perceive objects placed in high positions (e.g., shelves), while a drone is the best choice for this task. Given the current difficulty for the LLM model to generate novel algorithms, we write prompts to instruct the agent to assume the camera is symmetrical about the up-axis, check the angle $\alpha$ between the projected target object and the camera optical-axis, and compare it with half of the field-of-view (FOV) angle.

$$\theta_{FOV} = 2 \cdot tan^{-1}(\frac{x}{2f}) \tag{1}$$

- **Manipulation.** Robots feature diverse manipulation capabilities due to mechanical arms of different forms, various types of end effectors, and whether or not they have an explicit manipulator. Hense, it is important for agents to use mathematical tools to reason the robots arm workspace expecially when handling objects placed in abnormal positions (e.g. far inside the bed, high on the cabinet). For cooperative manipulation, the agents can pre-judge and assign the proper robots to fetch or place the target objects in the task planning stage.
which robots can reach the target objects.

These capabilities include both textual summarization and numerical details. While the capability summaries are used for common sense reasoning, the numbers are prompted to be used in LLM code generation for spatial-aware reasoning, which will be discussed in the next section.

## A.2   MULTI-AGENT SYSTEM DESIGN AND COMMUNICATION

**Hierarchical MAS Communication Graph** We design the MAS of EMOS following the HMAS-2 framework, which has been proven to be the most efficient LLM-based multi-robot communication

framework by Chen et al. (Chen et al., 2024). In particular, this MAS has one leader LLM agent for high-level task planning and subtask assignment, and several robot LLM agents to provide additional feedback back to the leader LLM agent given the assigned subtasks. The MAS communication graph can be referenced in figure 7, similar to a star topology.

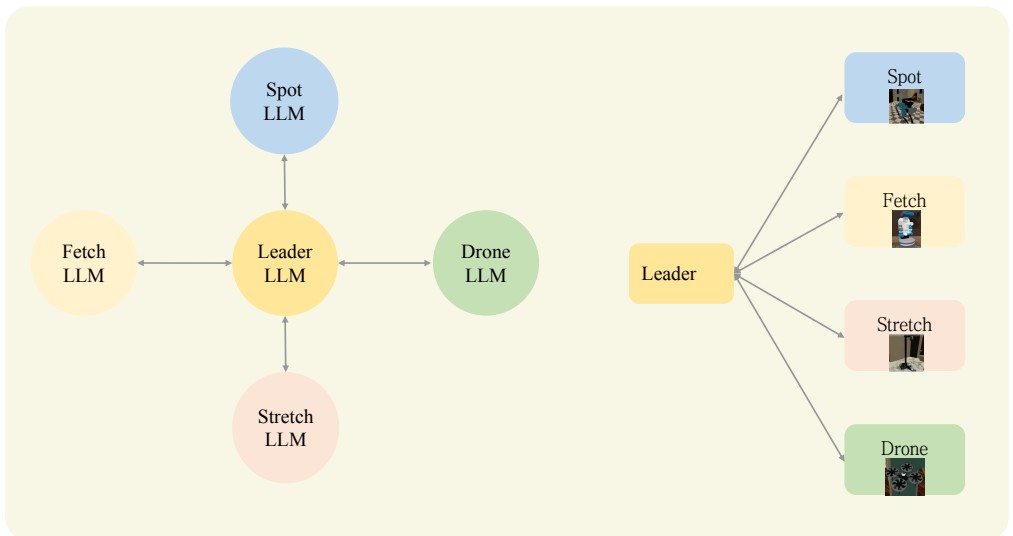

Figure 7: **Communication Graph Design for EMOS** This figure show our design on MAS communication graph of EMOS, leader LLM agent decompose the task and assign task to each robot LLM agent while robot LLM agents provide individual feedbacks for leader LLM agent to replan.

As we discussed in section 3.4 and algorithm 1, hierarchical task planning has 2 stages including **centralized group discussion** and **decentralized action execution**. The LLM agent leader will decompose the initial task into subtasks and assign them to different robot agents without embodiment reasoning, which can be referred to by the black arrows in figure 7. Then robot LLM agents will then reason with embodiment awareness through robot resumes themselves, judge whether they themselves can complete the given subtask according to its own embodied capability, and feed it back to the leader agent for replanning referring to the red arrows in figure 7. If each robot agent replies yes, then the task assignment will be executed in the following steps. A more detailed explanation of the two stages of the algorithm is provided.

**Centralized Group Discussion and Task Assignment** In the first stage of centralized group discussion, there is a `CentralPlanner` that generates an initial plan for each robot, and each robot also has an LLM agent that checks its assigned subtask and provides feedback to the central planner in `Reflection`. With the robot resume composed of general description and numerical details, the robot-dedicated agent could reason its general availability for the assigned task and further check the geometric availability by using mathematical tools. Specifically, by accessing robot resumes in the last section, a robot-dedicated agent is encouraged to perform **common-sense reasoning** with textual summaries, and generate code with numerical data in both the scene description and the robot resume to perform calculations for **spatial-aware reasoning**.

**Decentralized Action Execution** In the second stage, with the result of the assignment of tasks, each robot-dedicated agent starts to execute its action in parallel. Given a subtask description and action execution history, a robot-dedicated agent controls the current agent by LLM `FunctionCall` with robot control libraries. These robot control libraries are implemented with ground truth world information, classic robot trajectory planners and inverse kinematics solvers. An agent automatically goes to *wait* state when it finishes all reasoned actions. When the agent fails to accomplish a subtask but "believes" to have finished it, it could continue to execute the following planned actions. A side note is that, to study this circumstance of partial failure, we also evelute the task by sub-goals, which will be discussed in the experiments. The agent in the *wait* state will awake when it receives a new

task assignment from a group discussion. For an episode, the task ends when all robots are in the *wait* state.

## B    APPENDIX: ADDITIONAL DETAILS FOR HABITAT-MAS

### B.1    HABITAT-MAS BENCHMARK TASK DESIGN

As mentioned in the previous section, we carefully designed four challenging tasks to evaluate the embodiment-aware reasoning capabilities of MAS. Detailed description of each task is as follows:

- **Task 1: Cross-floor object navigation.** As an extension of the Multi-ON (Wani et al., 2020) problem, this multi-floor task requires the collaboration of robots with different base types to navigate to multiple objects in the scene. The wheeled robot can only operate on a single floor, while the legged robot can navigate between floors, emphasizing the need for coordinated planning with awareness of the mobility capabilities of different robots. This task is specifically designed to test the MAS's ability to reason about mobility constraints when coordinating cross-floor tasks.

- **Task 2: Cooperative perception for manipulation.** Due to limitations in perception caused by the camera's position and type, some articulated robots like spot may lack the ability to detect target objects on high shelves, while some arm-less robots like Drone with better camera view may succeed. The heterogeneous robots need to cooperate to acquire a good RGB-D perception of objects for precise manipulation. In this single-floor task, different target objects are placed in positions that are visible for certain robots. We aim to test whether the MAS can reason about the robot sensor type and viewpoint and successfully assign appropriate robots to perceive all target objects.

- **Task 3: Collaborative single-floor home rearrangement.** As we discussed in the last section, different robots have different manipulation capabilities. Articulated robots are limited to reaching objects within their arm's workspace. For instance, Stretch is equipped with an arm that can extend farther horizontally, while Fetch has a greater vertical reach, allowing it to grasp higher objects compared to Stretch. This single-floor task involves rearranging objects placed in varying positions, including the ground, high shelves, or a bed center far from a navigable area, which requires robots with different arm workspaces to understand their availability for different rearrangement targets.

- **Task 4: Multi-Robot, multi-object, multi-floor collaborative rearrangement.** This is a comprehensive task that requires complex coordination for collaboration. Within this scenario, several distinct types of robots, aerial, wheeled, and legged robots, must collaborate to perceive and rearrange a large set of objects distributed in various positions across multiple floors. This task combines the coordination of different capabilities of heterogeneous robots. Specially, some objects are located on high surfaces, such as cabinets upstairs, requiring advanced perception, manipulation and mobility. Our primary goal is to evaluate the MAS's ability to optimize task execution by effectively leveraging the unique capabilities of each robot, while balancing token efficiency and time step consumption.

### B.2    SUB-GOAL DEFINITION WITH PDDL LANGUAGE

Habitat environment Puig et al. (2023) has already integrated a PDDL system (McDermott et al., 1998) for the definition of composite tasks and the evaluation of the objectives in simulation. The goal of a complex task can be defined by a composite logical expression with primitive predicates and logical operators. Based on the PDDL system, we take the following primitive predicates as the sub-goals in evaluation: 1) **Robot_At_Object:** This is the first stage in every task, in which the robot needs to firstly navigate to the nearest navigable points to target objects and then execute the following actions like detect or pick. 2) **Robot_At_Receptacle:** This is another type of navigation sub-goal for long-horizon tasks like rearrangement ,for which robots need to navigate to the receptacles before placing objects on the recepcles to complete the final goals of rearrangement. 3) **Object_At_Receptacle:** This is the final goal of the rearrangement tasks. Sometimes robots may be assigned to pick and place objects beyond their reach, which means that it is not enough to just count whether robots can navigate to objects or receptacles. We add this sub-goal to test the ability to reason long-horizon tasks further explain which robot tend to fail in specific tasks. 4) **Robot_Detect_Object:** This is for specific perception tasks, aiming to judge how well the system performs in detecting objects.

### B.3 EPISODE GENERATION AND VERIFICATION

The episodic datasets in the benchmark are automatically generated by a cascaded sampling and verification process. We first fliter the eligible scenes from HSSD (Khanna* et al., 2023) and MP3D Chang et al. (2017) datasets that meet the defined requirements. For HSSD scenes, we ensure that the navigable points in the scene are connected and that there are a sufficient number of rooms available for placing receptacles and objects. Additionally, we calculate the navigable mesh(navmesh) and verify that the robot can navigate to the closest navigable point near the object. As for MP3D scenes, we ensure that the scene includes a multi-floor setup and that the different floors are connected by stairs which are steep enough to prevent wheeled robot from passing through but feasible for spot robot to navigate.

To sample an episode $E = (L, \mathbf{P}^0, T, G)$, we carefully select receptacles with specific height or width characteristics to accommodate different types of objects such as $L$, contributing to the layout diversity of the episodes. The robots' initial position and joint poses $\mathbf{P}^0$ are carefully initialized to ensure that the robot can complete the assigned tasks including navigation, perception and manipulation based on its capabilities. At the same time, the distance between the initial positions of the agents $R_i^0$, $R_j^0$, measured by Euclidean distance $d_{\text{euclidean}}(R_i^0, R_j^0)$, are set to avoid collisions because they are too close to each other. By carefully designing the goal state $G$ of each task and generating corresponding task descriptions $T$, we ensure that the task description includes all objects in the layout, thereby testing MAS comprehensive reasoning and planning capability for the goal state as thoroughly as possible. In mobility and perception task dataset, the length of the navigation path from the robots' initial position to target object is ensured not to be excessively long, which would lead to increased navigation time, nor too short, which could lead to one of the robots completing the navigation task too quickly and entering a prolonged waiting state, resulting in unnecessary token consumption. An episode would be trivial if the line connecting the robot's starting point and target object forms a straight line and completely coincides with the navigation path, the robot's navigation process appears mundane. Therefore, when validating the dataset, we ensure that the ratio of the geodesic distance $d_{\text{geodesic}}(R_i^0, O_j)$ between the initial position of the robot $R_i^0$ and the position of the target object $O_j$ to the Euclidean distance $d_{\text{euclidean}}(R_i^0, O_j)$ between the initial position of the robot and the object $P(R_i^0, O_j)$ defined in Equation 2 is greater than 1, while also ensuring that it is not excessively large to avoid an overly complex navigation path.

$$P(R_i^0, O_j) = \frac{d_{\text{geodesic}}(R_i^0, O_j)}{d_{\text{euclidean}}(R_i^0, O_j)} \tag{2}$$

To verify the episode is meaningful, i.e. to discriminate dummy policies and policy based on embodiment-aware reasoning, we introduce a set of validation criteria to 4 tasks, ensuring that the policy requiring multi-step planning and high-level perception outperforms random dummy policies. In particular:

**1)Navigation and multi-floor rearrange.** The object and receptacle in the navigation episode we filtered are not on the same floor. In the meantime, fetch robot and spot robot are initialize on the same floor, which means that if the MAS system assigns tasks correctly, spot robot must be assigned to perform cross-floor navigation or rearrange tasks, while the fetch robot(wheeled) can only be assigned to navigate to target objects on the same floor and iteract with them. Besides objects and receptacles placed on different floor in navigation episode, for objects located on the same floor as the robots' initial position, we excluded object that could be reached by spot robot, ensuring that they can only be operated by fetch robot, which further reduces the success rate in multi-floor rearrange task for dummy policy.

**2) Perception and single-floor rearrange.** For perception and single-floor manipulation tasks, we assume that we initially have the robot type settings with a pair of robots that are equipped with varying capacities within the same ability. We meticulously designed a comparative selection experiment, where robots with identical settings were tasked with completing opposite tasks using a fixed policy (a predefined task execution sequence, similar to an oracle plan). For example, in the positive task, the Fetch robot rearranges object1 and the Stretch robot rearranges object2, while in the negative task, the roles are reversed: the Stretch rearranges object1 and the Fetch rearranges object2. We filtered episodes where the success rates (0 or 1) in the positive and negative experiments were

XORed, thereby identifying episodes that can only be completed under specific arrangements. This approach aims to differentiate a dataset that cannot be completed using random (dummy) policies, to further demonstrate the effectiveness of our approach.

### B.4 DETAILED IMPLEMENTATION OF ROBOT LOW-LEVEL CONTROL IN BENCHMARK

In section 4.1, we generally introduce the robot low-level action implementation in our benchmark as we disable the physics simulation of integrated Pybullet Coumans & Bai (2016–2021). Here we will describe the implementation of each low-level action in detail with non-physics simulation and full observed settings.

For navigation action `navigate_to`, as we observed a target object like a apple on the table, the object's position in world coordinate framework can be calculated. Then we infer the path (series of waypoints) from the start position of the robot to the target position on navmesh through greedy search, and start to force the robot to move towards the next waypoint following the navigation algorithm used in Habitat Puig et al. (2023).

For arm action including `move_arm_to`, `pick`, `place`, since we have turned off the physical simulation including collision detection, the robotic arm control only requires that after the mobile robot observes the target objects, it first moves to a suitable position near the target position. Subsequently, based on the world coordinates of the object, the joint pose corresponding to the object coordinates is calculated through inverse kinematics, to achieve grasping using the suction grasp.

It should be noted that low-level control is not our concern, but our framework is easily extensive to introduce other control method like RL policy. Since action strategies are not directly related to embodied task planning, we did not discuss the underlying action strategies in great detail in the article.

## C APPENDIX: ADDITIONAL EXPERIMENTS

### C.1 SPECIAL CASE STUDY

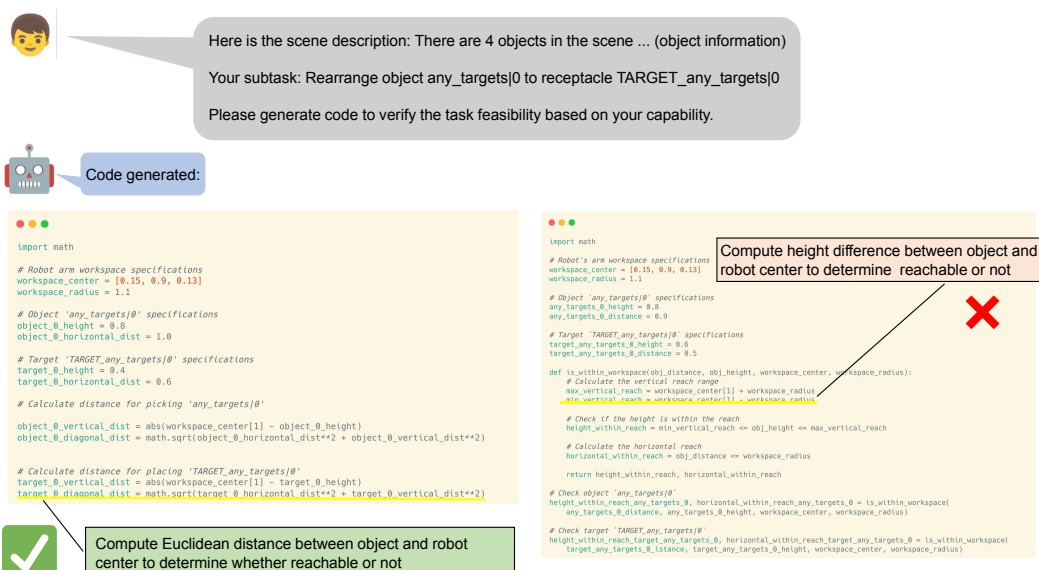

Figure 8: **Speicial case during agent reflection** This image illustrates the process of task inspection conducted by LLM agents during agent reflection. On the left side is the correctly generated code, while on the right side is an incorrect interpreration caused by hallucinations.

**Incorrect code generation caused by hallucinations** During large-scale experiments, we identified some specific cases that occasionally lead the system to make incorrect judgements about

the environment's state, particularly when the prompt contains numerical information, especially coordinates, even when we use structured text as the prompt for LLM. The code shown in Figure 8 demonstrates how LLM agents verify whether the robot can interact with the target object based on the object's height, horizontal distance between nearest navigable point from object to itself given scene description, robot height, and arm workspace (max reach of the robot arm) parsed from URDF. The correct verification method is to calculate whether the Euclidean distance from the robot's center to target object, when the robot is positioned at a navigable point, falls within the robot's reach. This is the code logic generated by LLM in most cases. However, there are still some cases where the large model fails to correctly understand the spatial relationships, even when we explicitly state through structured text that the calculation of whether the object is within the robot's reach should be based on horizontal distance and height difference. One common scenario is that the code on the right part only compares the height difference between robot and target object with the radius of the robot arm workspace. In other cases, the LLM agent may hallucinate a third coordinate beyond the horizontal distance and height difference to compute distance. Both of these cases can cause the robot to misjudge its range of manipulation, ultimately resulting in inappropriate task assignments.

## C.2 TOKEN ENHANCEMENT EFFICIENCY

Table 2: **The results of Token enhancement efficiency**

| Method | Relative Success Rate | Relative Token Usage | Token Efficiency $\downarrow$ |
|---|---|---|---|
| EMOS (Ours) | 1.00 | 1.00 | **2135.99** |
| w/o. Numerical | 1.61 | 1.52 | 2258.11 |
| w/o. Robot resume | 2.42 | 1.25 | 4133.08 |
| w/o. Discussion | 2.48 | 2.22 | 2388.51 |

EMOS consumes the most token since we include procedures such as leader assignment, group discussion, self reflection, action execution, etc. All of these procedures inevitably require a large number of LLM tokens. To evaluate the extra tokens consumed as we develop our framework resulting in a higher success rate, we define the relative success rate, relative token usage, and token efficiency. As shown in Table 2, to judge token efficiency, EMOS outperforms other ablated methods, when comparing the improvement in success rate and the increase in token usage, EMOS achieves a more increased success rate than token usage compared to all other ablated settings. To judge the token used per success rate, which reflects the tokens used to perform the tasks, EMOS performs the best in this metric. It shows that, as the token usage increase, our EMOS multi-agent system still perform the best in token efficiency.

## C.3 DISCUSSION ON OBSERVED PHENOMENA

Besides, there are some interesting phenomena: 1) EMOS w/o. group discussion beats EMOS w/o. robot resume in both navigation task success rate ($22.42\% \rightarrow 14.14\%$) and average sub-goal success rate ($72.45\% \rightarrow 65.27\%$): It indicates that multi-agent role playing might increase hallucination, especially when robots make decisions with overly complex description. 2) EMOS w/o. numerical performs just as badly as EMOS w/o. robot resume in complex tasks ($9.20\% \leftrightarrow 12.99\%$ in single-floor rearrangment): It suggests that LLM agents cannot actually reason about the manipulation limitation with URDF, indicating the importance of calling mathematical functions in our EMOS framework. 3) EMOS w/o. robot resume uses more tokens and steps than EMOS w/o. numerical: It reflects that even though the EMOS w/o. robot resume can consume fewer tokens with less textual input, the LLM agents must query involving more steps due to the low success rate in planning.

## C.4 EVALUATION ON SUB-GOAL SUCCESS RATE

As a supplement, Table 3 shows the sub-goal success rate of the five settings, providing a detailed insight into how well each setting performs in planning and execution of sub-tasks. All settings perform relatively well in the Robot_At (the first and the second) sub-goals, which just require the robots to navigate to the target points. By comparing EMOS w/o. robot resume with other settings,

Table 3: **Sub-goal success rate of EMOS and ablated methods on Habitat-MAS benchmark.**

| Sub-goals | EMOS (Ours) | w/o. Numerical | w/o. Robot resume | w/o. Discussion |
|---|---|---|---|---|
| Robot_At_Object↑ | **88.27%** | 77.48% | 69.48% | 80.59% |
| Robot_At_Receptacle↑ | **88.90%** | 80.63% | 76.15% | 83.56% |
| Object_At_Receptacle↑ | **57.04%** | 40.38% | 40.95% | 47.70% |
| Robot_Detect_Object↑ | 93.81% | **97.35%** | 79.57% | 70.80% |

as we discussed before, due to not having the summary of mobility description, it can be inferred that robots in this setting tend to fail in multi-floor navigation tasks. It is also notable that compared to EMOS w/o. numerical and EMOS w/o. robot resume, EMOS w/o. group discussion achieves higher sub-goal success rates in navigation sub-goals, while lower in detection sub-goals. This reflect that in this setting, MAS tend to fail in the tasks that require robot to collaborate to complete, which affirm the superiority of the group discussion in our framework, which can indeed reduce the homogenization in multi-agent task planning.

## C.5    EXTRA EXPERIMENT ON FORMAT OF ROBOT RESUME

In order to study which form of robot resume the LLM uses for reasoning with the highest efficiency, we conducted experiments among different formats of robot resumes (natural language, JSON, markdown and XML) for embodied task planning, which are generated using GPT-4o.

Specifically, we sample 10 episodes from the perception task and evaluate the average success rate of each format. The experimental results can be referred to Table 4

Table 4: **Average Success Rate Using Different Format of Robot Resume**

| Format | Natural Language | JSON | Markdown | XML |
|---|---|---|---|---|
| Avg. Succ Rate | 0.3 | **0.7** | 0.5 | 0.6 |

Our experiments reveal that structured formats, such as JSON and XML, outperform unstructured formats like natural language in achieving higher success rates for robot resumes. Notably, the success rate increases as the format becomes more structured, which aligns with a key observation in recent research on large language models (LLMs). However, during the experiments, we observed that certain formats, such as Markdown and XML, can induce hallucinations in LLMs. In these cases, the agents do not generate the properly formatted actions, resulting in a 0% success rate. To enable meaningful comparisons, we refined the prompts with minimal modifications to address this issue and produce usable results.

Based on the observations, the JSON format, used in our frameworks, performs the best. To answer the question about how to generate a better-formatted resume, we suggest using structured formats like JSON format in our frameworks, rather than loosely structured formats like natural language.

## C.6    EXTRA EXPERIMENTS ON SCALABILITY OF EMOS

Scalability is a crucial aspect of designing a robust framework like EMOS, as it enables seamless integration of new features and ensures that the system can adapt to evolving requirements. While the core functionalities of EMOS have demonstrated effectiveness in Habitat-MAS, we also provide additional experiments to evaluate how well the framework accommodates new capabilities and scales across different tasks and conditions. By testing these attributes, we aim to provide some insight about the scalability of EMOS when extending its scale, which could help the community working on scalable multi-agent systems to understand the system characteristics better.

**Scalability with Robot Number** In order to further verify whether EMOS can be applied to systems with increasing number of robots, robot types or task complexity to test the scalability, we conduct experiments by scaling the number of robots performing the same task. In our experiments, we sample 10 episodes from the manipulation task and evaluate performance across different numbers of Fetch robots (2, 4, 6, and 10) to assess communication efficiency and success rate.

As shown in table 5, we found that as the number scales up, the multi-agent system will face problems like hallucinations (in the setting of 10 agents) and the average success rate will decline. This is as expected since the hallucination problem in LLM is prevalent and it becomes worse with the increase of context length. On one hand, this could be alleviated with more powerful LLM models as we have witnessed in the recent progress of LLM models. In the other hand, designs like hierarchical communication with smaller subgroup discussions and larger group aggregation (similar to delegate meeting) could help solve the scalability problem in multi-agent discussion.

Table 5: Scaling up with Robot Number

| Num of Robots | Succ. Rate | Token Usage |
| --- | --- | --- |
| 2 | 80% | 48779 |
| 4 | 60% | 73202 |
| 6 | 70% | 93252 |
| 10 | 50% | 151952 |

Table 6: Scaling up with Object Number

| Num of Objects | Succ. Rate | Token Usage |
| --- | --- | --- |
| 1 | 90% | 25778 |
| 2 | 80% | 50005 |
| 3 | 80% | 87668 |
| 5 | 70% | 197485 |

**Scalability with Object Number** Considering the scalability problem from another perspective, we conduct a second experiment focusing on increasing task complexity by scaling up the number of objects to manipulate, referring to table 6. While maintaining the fixed number of Fetch robots of (2), we evaluate the system's performance with varying numbers of objects (1, 2, 3, and 5).

In the second experiment, which involved scaling up task complexity, we found that our system demonstrates robustness to a certain extent. As the number of objects increases — indicating greater task complexity — the system maintains a relatively high and stable success rate (above 70%). However, the increasing hallucination problem still exists under this setting.

