# OpenReview forum: "EMOS: Embodiment-aware Heterogeneous Multi-robot Operating System with LLM Agents"
_ICLR.cc/2025/Conference — ICLR 2025 Poster_

### Official Review · Reviewer_xZUU · 2024-10-19

**Soundness:** 2
**Presentation:** 2
**Contribution:** 2
**Rating:** 5
**Confidence:** 4

**Summary:**

This paper proposed a heterogeneous multi-robot system framework to enable effective collaboration among heterogeneous robots with varying embodiments and capabilities. However, from my perspective, the topic of this paper is too big, making it more like a technical instruction or report rather than a research paper. Moreover,  many details are not clearly explained, such as low-level planning and control (collision avoiding), mid-level decision-making, and high-level learning. Moreover, the communication graph is an important part of the MAS design, which is also not mentioned in the framework. I suggest the authors reorganize the paper, abstract the model, and then discuss it from theoretical and practical perspectives.

**Strengths:**

This paper introduced a heterogeneous multi-robot system framework to enable effective collaboration among heterogeneous robots with varying embodiments and capabilities. It implements the proposed LLM-based multi-agent framework in indoor household environments with four tasks. The paper presents the LLM-based MAS framework conducting embodiment-aware reasoning in simulation.

**Weaknesses:**

From my perspective, the topic of this paper is too big, making it more like a technical instruction or report rather than a research paper. Many details are not clearly explained, such as low-level planning and control (collision avoiding), mid-level decision-making, and high-level learning. Moreover, the communication graph is an important part of the MAS design, which is also not mentioned in the framework. Furthermore, several robots' plannings have apparent errors in the simulation video. For example, at the 16 seconds of the video, the drone does not follow the routine but goes through the wall directly, etc. I hope the authors can improve it and consider more details about different modules in the framework design.

**Questions:**

I suggest the authors reorganize the paper, abstract the model and ideas, and then discuss it from theoretical and practical perspectives,
such as how to formularize the capabilities of heterogeneous multi-robot systems to satisfy various tasks' requirements, how to define the specific roles and relationships in the group, how to organize their behaviors or strategies to optimize system performances, etc.

I suggest the authors check and refer the paper as below:

1) Yang, Q., & Parasuraman, R. Hierarchical needs based self-adaptive framework for cooperative multi-robot system. In 2020 ieee international conference on systems, man, and cybernetics (smc) (pp. 2991–2998). IEEE.

2) H. Hamann and H. Wo ̈rn, “A framework of space–time continuous models for algorithm design in swarm robotics,” Swarm Intelligence, vol. 2, no. 2-4, pp. 209–239, 2008.

---

> ### Author Response · Authors · 2024-11-20
> **Rebuttal Comment**
>
> We really appreciate the reviewer for the effort in review and advice. And we do apologize for not clarifying the papers’ focus enough.
>
> ### **Regarding to Detailed Discription**
>
> Although we have mentioned a big topic in this paper, as we described in the introduction (section 1, line 100), our focus is the _embodiment-aware task planning_ using the LLM multi-agent methodology, which is related to the high-level task planning.  For detailed high-level decision making process in our framework, we have illustrated in section 3.4, line 302.
>
> We do not prominently emphasize the contribution of the entire system in low-level motion planning, as we have mentioned in section 4.1, line 390, the collision has been disabled with Pybullet physics simulation in this benchmark.
> Besides, we acknowledge the detailed description is not discussed enough. Thus we add section _B.4 Detailed Implementation of Robot Low-level Control in Benchmark_ for clarity.
>
> ### **Regarding to Communication Graph**
> We really thank you for your suggestion. We discribe the communication graph and discussion process in section 3.4 but without visualization.
> And we have added the communication graph in the appendix _A.2 MAS Communication Graph_ for graph visualization and detailed discription to address this concern.
>
> ### **Regarding to Error in Video**
> We are really sorry for the confusion caused by the video. For the drone at the 16th second in the video, in fact the drone passed through the room door following the routine, which can be seen from this screenshot that drone passes through the door. Please refer to this picture [CLICK](https://github.com/EMOS-project/EMOS-project.github.io/blob/main/rebuttal/drone_pass_door_bbox.png), in which the door that drone flying through can be clearly seen. We’ve rechecked the demo videos on the website and confirmed that all the robots in the video behave correctly.
>
> We sincerely express our appreciation for reviewer's valuable advice, and we'll update the draft according to those suggestions and questions as soon as possible. We do hope the refinement and explaination can win our reviewer back and look forward to further discussion.

---

> ### Author Response · Authors · 2024-11-23
> **Kind reminder for review response**
>
> Dear reviewer,
>
> We hope this message finds you well. We are writing to request your response to our rebuttal comment kindly. We would appreciate it if you could review our response and we could have helpful discussions here.
>
> We have revised the draft as you suggested:
> - We reorganized the `Section 2. Related Works` and their relations to our paper as suggested by **Reviewer jjgY** and **Reviewer xZUU**.
> - We added `Appendix A.2 MULTI-AGENT SYSTEM DESIGN AND COMMUNICATION` according to the request by **Reviewer xZUU** to add the agents' communication graph
> - We added `Appendix B.4 DETAILED IMPLEMENTATION OF ROBOT LOW-LEVEL CONTROL IN BENCHMARK` according to the request by **Reviewer xZUU**.
>
> Please let us know if you have any more questions and we are happy for further discussions.
>
> Kind regards,
>
> The authors

---

> > ### Author Response · Authors · 2024-11-26
> > **Second kind reminder for response**
> >
> > Dear reviewer,
> >
> > We hope this message finds you well. This message kindly reminds you that the deadline for the authors to upload the revised paper is November 27th. Please let us know your ideas about the current revision and further questions/ suggestions if any.
> >
> > We understand you are quite busy these days due to the heavy workload of rebuttal and discussions. We hope you are satisfied with our latest revision, in which we tried to address three suggestions in your review. We will also be happy to hear from you about further revision suggestions/ discussions.
> >
> > Kind regards,
> >
> > The authors

---

> > > ### Author Response · Authors · 2024-12-01
> > > **Third kind reminder for response**
> > >
> > > Dear reviewer,
> > >
> > > We hope this message finds you well. This message kindly reminds you that the **deadline for the reviewer to respond is Dec. 2nd**. Please let us know your questions or suggestions, if any.
> > >
> > > We understand you are quite busy these days due to the heavy workload of rebuttals and discussions. We hope you are satisfied with our revision to address the three questions/ suggestions in your review.
> > >
> > > Since our last reminder, we also have the following updates on the draft:
> > > - We added `C.6 EXTRA EXPERIMENTS ON SCALABILITY OF EMOS`. This section discusses the scalability of the EMOS framework.
> > >
> > > We are looking forward to hearing from you about further revision suggestions and discussions.
> > >
> > > Kind regards,
> > >
> > > The authors

---

### Official Review · Reviewer_jjgY · 2024-10-31

**Soundness:** 3
**Presentation:** 3
**Contribution:** 3
**Rating:** 8
**Confidence:** 5

**Summary:**

The paper presents an interesting approach to LLM-based multi-robot systems where robot roles are not predefined but are instead determined dynamically based on the robots' physical capabilities. The authors introduce the concept of a "robot resume," generated from robot URDF files and kinematic data, to guide task planning and integrate role flexibility within the model.

This concept is very compelling; however, my main concern is the additional degree(s) of freedom introduced into the decision-making and planning. In systems with predefined/assumed heterogeneous roles, the answer to “who does what” is partially predetermined, simplifying the planning process. Here, this isn’t the case, and the current centralized hierarchical LLM-based planning and task assignment system becomes significantly constrained. With the current design, the centralized planner and task assignment modules will face considerable burden in optimizing plans and assignments, particularly as domain complexity increases or when dealing with similar or more universally capable robots. I suggest comparing your approach against systems with predefined roles to illustrate any potential advantages or limitations. Additionally, you can provide a quantitative analysis of how your system's performance scales with increasing numbers of robots, robot classes, or task complexity (see [1] for example).

[1] Seraj et al. "Learning efficient diverse communication for cooperative heterogeneous teaming", AAMAS 2022

One potential solution to mitigate this issue could involve incorporating prior knowledge to streamline planning and decision-making. For example, a standard quadcopter lacks object manipulation capabilities, so it can be automatically excluded from relevant tasks, reducing the computational load. Such prior knowledge could be represented as a context/class vector for a contextual classification. Additionally, other factors such as robot availability (i.e., whether a robot is still executing a prior task) could contribute to a truly asynchronous multi-robot system.

I also have a few questions:
- Can you discuss the modularity of this approach? For instance, can you describe the process for adding a new capability like battery life to the robot resume and how it would be integrated into the task planning and allocation process. This would give a clearer picture of the system's modularity and extensibility.
- Do LLM-based planners consider assignment risks, and if so, how can these risks be leveraged in the decision-making process?
- It remains unclear how the central planner optimizes "which robot does what." For example, how does the system handle scenarios with (1) multiple equally capable robots, (2) differently capable robots that can all execute a task, or (3) universally capable robots? Does the LLM-based “discussion” incorporate strategic negotiation or cognitive hierarchy mechanisms (e.g., k-level thinking) to optimize plans and assignments? Can you provide a specific example scenario for each of these cases and explain how the system would handle the allocation in each case?
- Given the dynamic nature of the environment, assuming full observability is unrealistic. However, it is also unclear how partial observability is handled in this system. Can you discuss how your system might be adapted to handle partial observability, and what challenges you anticipate in doing so?

Additionally, I suggest including a section on Heterogeneous Multi-Robot Systems literature rather than the current Multi-Robot Systems section. Given the context of this work, it would be more impactful to emphasize on the heterogeneity in collaborative multi-robot teams, categorizing and introducing previous approaches such as MARL (heterogeneous multi-robot communication and coordination), Multi-Agent Apprenticeship Learning (learning heterogeneous multi-robot policies from human demonstrations), trait-based heterogenous multi-robot planning, and other non-LLM-based strategies. Below are a list of papers to begin with:

[1] "Learning efficient diverse communication for cooperative heterogeneous teaming", AAMAS 2022

[2] "Heterogeneous Multi-Robot Reinforcement Learning", AAMAS 2022

[3] "STRATA: A Unified Framework for Task Assignments in Large Teams of Heterogeneous Agents", JAAMAS 2020

[4] "Mixed-initiative multiagent apprenticeship learning for human training of robot teams", NeurIPS 2023

[5] "Heterogeneous policy networks for composite robot team communication and coordination", T-RO 2024

Lastly, a minor but important point: the paper contains numerous grammatical and verbal errors. A thorough review is recommended.

Overall, I'm happy with the contributions made in this paper and vote for a weak accept. I'd be happy to raise my score given mine and my fellow reviewers' comments are adequately addressed. Thanks!

**Strengths:**

-- See above.

**Weaknesses:**

-- See above

**Questions:**

-- See above

---

> ### Author Response · Authors · 2024-11-20
> **Official Comment by Authors (part 1)**
>
> We sincerely appreciate the reviewer's thorough and insightful feedback on our paper. The reviewer has provided valuable comments on the strengths of our approach, particularly highlighting the innovative concept of "robot resume" for dynamic role determination in LLM-based multi-robot systems. We are grateful for their recognition of the compelling nature of our work. Since there are a lot of questions and discussions from the reviewer’s feedback, and the number of characters in a comment box is limited, we will respond in multiple parts:
>
> ### **Concern about degree of freedom for the system**
>
> We appreciate the reviewer's insightful question about the additional degrees of freedom introduced in our decision-making and planning process. Specifically, we agree with the reviewer’s point of “ ’who does what’ is partially predetermined, simplifying the planning”. In fact, we follow the conclusion in [1] which explored four different LLM multi-agent architectures for group discussion. Their conclusion is that the centralized decision making process greatly helps all agents to make agreement and the discussion “converges” to a concrete plan, i.e. sequence of sub-tasks and task assignment. For a constrained-free LLM multi-agent discussion, the system has a high probability of divergence, in which no agreement can be made or the discussion falls into a loop or mode collapse. We do acknowledge the limitation of this constrained task planning here, trading a bit of the freedom for the convergence and performance.  We will add the section of “Acknowledgment for Limitation” to discuss the potential limitations in the design for the extensive systems.
>
> ### **Suggestion on comparing our approach with other methods with predefined roles**
>
> We are really sorry that the writeup has led to some confusion. In the ablation study itself, we have already compared our framework with other similar baseline methods. For example,  w/o. Robot resume refers to Meta GPT[2] and w/o. Discussion refers to CMAS[1]. It is worth mentioning that Meta GPT[2] is a  role-playing method that assigns different roles (managers, developers, test engineers) manually to LLM agents as we discussed in the section 4.4 in the submission, lines [475 - 478]. Our method’s success rate, sub-goal success rate, and time efficiency all constantly outperform the role-playing method with human-created profiles. We really thank the reviewers for pointing out this ambiguity, and we will revise the draft as soon as possible.
>
> ### **Question of modularity and experiment on adding battery life**
>
> Thank you for your thoughtful question about the modularity and extensibility of our system. Our implementation leverages modularity through a `RobotResume` Python class, which stores robot capabilities in a JSON file. Different capabilities are inferred via separate modules, such as manipulation, perception, and navigation, each utilizing specific information like URDF files and camera parameters. To add new capabilities, such as battery life, we update the `defaults.py` file with the relevant descriptions and regenerate the robot resumes in JSON format. The new capability can be integrated into task planning by refining the agents' prompts to focus on the added feature, demonstrating the modularity and extensibility of our system.
>
> At the reviewer's request, and to also better illustrate the extensibility of the robot resume, we conducted experiments by adding battery life designed based on real-world conditions as a new capability. We tested the system on 5 episodes of perception tasks and 5 episodes of manipulation tasks. The results showed a 100% success rate in avoiding the allocation of robots whose task execution time would exceed their battery life, demonstrating the effective integration of the new capability into the task planning and allocation process. The result is expected thanks to the strong prior of common sense reasoning stored in LLMs. The chat history of the experiment is uploaded to our [repository](https://github.com/EMOS-project/EMOS-project.github.io/blob/main/rebuttal/exp_battery_life.zip)
>
> Considering the limited time for the rebuttal and limited comment length, this just tries to address part of the concerns and questions. We will upload the updated draft and the rest parts of the rebuttal comments as soon as possible. We really thank the reviewer's understanding.
>
> [1] Chen, Yongchao, Jacob Arkin, Yang Zhang, Nicholas Roy, and Chuchu Fan. "Scalable multi-robot collaboration with large language models: Centralized or decentralized systems?." In 2024 IEEE International Conference on Robotics and Automation (ICRA), pp. 4311-4317. IEEE, 2024.
>
> [2] Sirui Hong, undefined., et al, "MetaGPT: Meta Programming for A Multi-Agent Collaborative Framework," in The Twelfth International Conference on Learning Representations, 2024.

---

> > ### Author Response · Authors · 2024-11-21
> > **Official Comment by Authors (part 2)**
> >
> > ### **Risk consideration in planning and decision-making process**
> >
> > We do think this is a good question and we did not cover the discussions from this perspective in the main paper.
> >
> > In short, we don’t explicitly intervene with risk management, instead, the central planner conducts task assignments and avoids assignment risk through the reasoning ability of LLMs based on implicit hints. The LLM planner will implicitly consider the potential risks based on its common sense. We set some limitations in the prompt to ensure that the LLM-based planner will consider all robot agent individuals when performing task planning and ensure that all subtasks are assigned to a certain robot. During this process, it is possible that the same robot may be assigned multiple subtasks, or a certain robot may not be assigned any task, which are allowed and not regarded as risks. In the subsequent group discussion process, the subtasks are re-assigned through the reflection of robot agents, and it’s also used to check that there is no such risk that the same subtask is assigned to multiple agents. If an assignment risk occurs, the discussion will be repeated until the risk is eliminated.
> >
> > However, our work focuses more on heterogeneous multi-robot collaboration, aiming to propose the concept of embodiment-aware multi-agent task planning.  We do appreciate the reviewer’s valuable question, and we will discuss the limitations of our work on this aspect and the potential improvement towards enhanced risk-aware planning and decision-making.

---

> > > ### Author Response · Authors · 2024-11-21
> > > **Official Comment by Authors (part 3)**
> > >
> > > ## **Questions on how the system conducts task assignment**
> > >
> > > We are amazed by how detailed and thoughtful this question is, and we are grateful to the reviewer for his/her time and dedication to this review. Considering there are multiple sub-questions contained in this question, we will address them in sequence:
> > >
> > > ### **How the central planner optimizes task assignment**
> > >
> > > In this question, the reviewer asks about how our system handles task assignment under three scenarios with (1) multiple equally capable robots, (2) differently capable robots that can all execute a task, or (3) universally capable robots.
> > >
> > > We apologize for not clearly explaining the central planner's optimizing process. Since the central agent is LLM-based, the reason why we set up a benchmark consisting of 4 tasks is also to test the LLM’s ability in task assignment on the one hand. We optimize the task assignment through iterative group discussions and add some limitations to the prompt. Regarding the special cases mentioned by the reviewer, it relies more on the LLM’s reasoning ability on the robot resume and environment context. Our optimization approach is to set limitations through the prompt and add sufficient information in the robot resume and environment context to assist the reasoning of the central planner.
> > >
> > > ### **Does the LLM-based “discussion” incorporate strategic negotiation or cognitive hierarchy mechanisms**
> > >
> > > Our discussion does not involve traditional strategic negotiation or cognitive hierarchy mechanisms. As mentioned above, we guide the central planner to make correct judgments through some rule limitations and improve the assignment of the central planner through the embodied reasoning of the robot agent. It is a hierarchical structure similar to a 2-level thinking. The central planner has the highest level of cognition and decomposes and assigns the global tasks, while the robot agent only reflects on the tasks assigned to it in combination with the robot resume.
> > >
> > > ### **Can you provide specific examples for the scenarios**
> > >
> > > These three special cases can be simplified as follows: Multiple robots are capable of executing the same subtask. Our current system optimizes this situation in the following way: When multiple robots can execute the same subtask, the optimization goal changes to require the lowest energy cost for the entire system. We introduce information such as the positions of each object and robot in the environment context and other attributes like battery life in the robot resume. The central planner can first calculate information such as the geodesic distance between each robot and the target object through function calls. Finally, it calculates the energy cost for each robot to execute this subtask. Then, the central planner assigns the task according to the calculation results.
> > >
> > > Next, we will give a specific example for each special case.
> > >
> > > - For multiple equally capable robots, for example, 2 spots in the multi-floor navigation tasks that have objects on different floors to be rearranged, we will explicitly provide the current position and target position of each spot. LLMs can judge the distance cost for each spot to reach the goal position. Through group discussion, our system will assign the nearest robot capable of completing the task to execute. For example, if one spot is on the first floor, while the other is on the second floor, and the target object is on the first floor, then the central planner will assign the navigation task to the spot on the first floor to minimize the energy cost. The task is to navigate to a cabinet on the second floor. Then the latter spot will be assigned.
> > > - For differently capable robots, if all the differently capable robots can execute the same task, the LLMs will first consider the task cost just like we mentioned above, and then assign different robots to complete the tasks simultaneously. For example, if both the fetch and stretch can reach the objects placed in the house,  and the fetch is closer in geodesic distance to the target object than the stretch, then the central planner will assign the rearrange task to fetch to pick the nearest objects simultaneously for efficiency.
> > > - The situation of a universally capable robot is similar to the first case.
> > >
> > > We really thank the reviewer for your question and we hope these discussions help answer your questions.

---

> ### Author Response · Authors · 2024-11-21
> **Official Comment by Authors (part 4)**
>
> ### **Question on how the system handles partial observable environment**
>
> We thank the reviewer for this visionary question. Firstly, we admit that we assume the multi-robot system is equipped with a perfect multi-agent SLAM system, providing a full observability of the environment, as we mentioned in section 3.1, lines 203-206. While our current implementation focuses on fully observable scenarios, we also recognize the need to address partial observability for real-world applications. For this purpose, we propose several extensions that could extend our current system to handle partial observability:
> - Probabilistic State Estimation: We could incorporate probabilistic methods like Bayesian filtering or particle filters to estimate the full state based on partial observations. This would allow agents to reason about uncertainty in their knowledge of the environment.
> - Active Perception: We could incorporate active perception strategies, where agents actively seek information to reduce uncertainty about critical aspects of the environment.
> - Belief-Space Planning: Instead of planning in the state space, we could adapt our planning algorithms to operate in the belief space, accounting for uncertainty in the current state and future outcomes of actions.
>
> There are two challenges for this adaptation we think matter in this dynamic decision process. Firstly, agents need to synchronize observation with global representation to update the state, and this could be very costly and slow with LLMs. There could be some classical distributed SLAM algorithm to reduce this overhead. Secondly, reasoning about partial observability typically increases the computational burden and instability, which may require optimizations to maintain real-time performance.
>
> ### **Scalability of the framework**
>
> We recognize the importance of scalability of a multi-agent system and we are grateful to the reviewer for this question. Now we are emergently adding a new experiment to study the system's performance with the increase of robot numbers in the environment. We have started these two experiments:
>
> - In the first experiment, we are scaling the number of robots performing the same task. In our experiments, we sample 10 episodes from the manipulation task and evaluate performance across different numbers of Fetch robots (2, 4, 6, 10, and 20) to assess communication efficiency and success rate. However, due to computational limitations and the limited time of the rebuttal session, this experiment is still ongoing.
> - In the second experiment, we plan to study the impact of increasing task complexity by scaling up the number of objects to manipulate. While maintaining a fixed number of Fetch robots (2), we evaluate the system's performance with varying numbers of objects (1, 2, 3, 5, and 10).
>
> Some existing results from the first ongoing experiment indicate that, with the key design of leader allocation and group discussion, the system can successfully assign a minimal yet sufficient number of robots to complete the manipulation tasks.
>
> We will report the experiment results with analysis in the revised paper and comment boxes as soon as possible.

---

> > ### Author Response · Authors · 2024-11-25
> > **Update on Scalability Experiments' Results**
> >
> > We really thank the reviewer's insightful question on the scalability of the current framework. As promised before, we are now updating the results of the scalability experiments:
> >
> > ### **Table 1: Scalability with Increasing Agent Numbers**
> >
> > | Number of Robots | Success Rate (%) | Token Usage |
> > |-------------------|------------------|-------------|
> > | 2                | 80%             | 48779       |
> > | 4                | 60%             | 73202       |
> > | 6                | 70%             | 93252       |
> > | 10               | 50%             | 151952      |
> >
> > In the first experiments of scaling up the robot number, we found that as the number scales up, the multi-agent system will face problems like hallucinations (in the setting of 10 agents) and the average success rate will decline. This is as expected since the hallucination problem in LLM is prevalent and it becomes worse with the increase of context length. This could be alleviated with more powerful LLM models as we have witnessed amazing progress in LLM model capability in the past year. Or designs like hierarchical communication with smaller sub-group discussions and larger group aggregation (delegate meeting) could help to solve the scalability problem in multi-agent discussion.
> >
> >
> > ### **Table 2: Scalability with Increasing Task Complexity**
> >
> > | Number of Objects | Success Rate (%) | Token Usage |
> > |--------------------|------------------|-------------|
> > | 1                 | 90%             | 25778       |
> > | 2                 | 80%             | 50005       |
> > | 3                 | 80%             | 87668       |
> > | 5                 | 70%             | 197485      |
> >
> > In the second experiment, which involved scaling up task complexity, we found that our system demonstrates robustness to a certain extent. As the number of objects increases—representing greater task complexity—the system maintains a relatively high and stable success rate (above 70%). This indicates that our communication structure is effective and scalable, capable of handling more complex problems.
> >
> > For the experiments raw data, you can check this file: [exp_scalability.zip](https://github.com/EMOS-project/EMOS-project.github.io/blob/main/rebuttal/exp_scalability.zip)
> >
> > ### **Explanation of changing experiment setting**
> > We are conducting this experiment on a workstation with a single RTX 4090. Due to the limitations of the simulator and computational constraints, rendering camera sensors for 20 robots always caused the program to crash. If time permits, there might be a workaround for the simulation to reduce the resource consumption. As a result, we simplified the experimental settings and removed the setting for 20 robots.
> >
> > We thank you again for your thoughtful question.

---

> ### Author Response · Authors · 2024-11-21
> **Official Comment by Authors (part 5)**
>
> ### **Suggestion on using prior knowledge to improve planning and decision-making**
>
> We appreciate the reviewer's insightful suggestion about incorporating prior knowledge to improve planning efficiency. We want to clarify that our EMOS framework already implements this idea through our novel "robot resume" approach (Sec 3.3), but in a more comprehensive and flexible way than using simple context/class vectors. Here's why:
>
> 1. Hardware-specific capabilities: Our robot resume captures each robot's physical capabilities by analyzing their URDF files and using forward kinematics to generate numerical specifications. For example, for a quadcopter, its resume would indicate no manipulation capability, automatically excluding it from relevant tasks.
> 2. Context representation: The robot resume provides both textual summaries for common-sense reasoning and numerical specifications for precise spatial calculations. This dual representation allows agents to quickly filter out infeasible tasks while maintaining the ability to perform detailed geometric verification when needed.
> 3. Handling complex tasks: Simple context vectors would struggle with complex, compositional tasks. For example, consider a task like "Find the toy in the bedroom on the second floor and place it on the high shelf in the living room." While a context vector might indicate a robot has manipulation capability, our resume-based approach can:
>     - Use numerical workspace analysis to verify if the robot arm can reach the high shelf height
>     - Check mobility constraints for multi-floor navigation
>     - Consider perception capabilities to ensure the robot can visually locate the toy
>     This level of detailed capability matching would be difficult to encode in simple class vectors.
>
> Regarding robot availability, while our current implementation focuses on physical capabilities, the framework can be naturally extended to include dynamic state information (e.g., current task status) in the robot resume.
>
> Finally, we really appreciate the reviewer for the careful review of this paper, and we will correct the grammar errors in the paper and update the revised version of the paper as soon as possible.

---

> ### Author Response · Authors · 2024-11-23
> **Kind reminder for review response**
>
> Dear reviewer,
>
> We hope this message finds you well. We are writing to request your response to our rebuttal comment kindly. We would appreciate it if you could review our response and we could have helpful discussions here.
>
> We have revised the draft as you suggested:
> - We reorganized the `Section 2. Related Works` and their relations to our paper as suggested by **Reviewer jjgY** and **Reviewer xZUU**. Specifically, we divided the "Multi-Agent System" subsection into two sections "LLM-Based Multi-Agent System" and "Heterogeneous Multi-Agent Learning".  All the mentioned related works by these two reviewers have been added to the reference list.
>
> We are looking forward to your responses and we also welcome further discussions.
>
> Kind regards,
>
> The authors

---

> > ### Comment · Reviewer_jjgY · 2024-11-25
> > **Response to Authors**
> >
> > I really appreciate the thorough discussions, additional experiments, and mindful clarifications provided by the authors in their rebuttals. I increased my score respectively as I believe both my understanding of the contributions and the paper itself both are in much better shapes now.
> >
> > Good Luck.

---

> > > ### Author Response · Authors · 2024-11-25
> > > **Reply to Reviewer jjgY**
> > >
> > > We do appreciate the reviewer's insightful suggestions and taking the time to engage deeply with our work. The suggestions from the reviewer greatly help us improve the paper. Also, we are grateful that our efforts and dedication are recognized by the reviewer.

---

### Official Review · Reviewer_79K2 · 2024-11-02

**Soundness:** 3
**Presentation:** 4
**Contribution:** 3
**Rating:** 8
**Confidence:** 3

**Summary:**

This paper introduces a novel framework for controlling heterogeneous multi-agent collaboration, named EMOS. This framework integrates recent advancements in large language models (LLMs) and multimodal models, enabling intelligent cooperation among multiple robots. Unlike previous works, which primarily relied on role-playing approaches, this multi-agent framework innovatively proposes a robot resume based on the robot's URDF files. This enhancement allows for a more precise description of the robots' capabilities, thereby facilitating more efficient communication and collaboration among them. Additionally, the authors present a new benchmark named Habitat-MAS. This benchmark validates the framework's effectiveness. The paper also provides several specific experimental results, which have been published on a dedicated website.

**Strengths:**

Robots’ resumes are generated by the LLM based on urdf files, which enhance the communication among robots. This paper more effectively leverages the intelligence of LLMs in addressing the challenges of heterogeneous multi-robot collaboration. LLM can not only play a planner role but also promote the communication effectiveness among robots.

**Weaknesses:**

The experimental section of the paper requires additional enhancements. While the authors conducted ablation studies on their framework and provided detailed results, there is a notable absence of comparisons with similar frameworks.

**Questions:**

1）In the domain of heterogeneous multi-agents, how does the proposed framework's use of robot resumes improve performance compared to traditional role-playing methods? Moreover, does the framework consistently outperform others across various scenarios and different robot configurations?
2）All robot are completely interconnected, and they are collaborative under complete information. Can the proposed framework be extended to scenarios where there is a game relationship between heterogeneous individuals.
3)  What is the scalability of the framework? How does the increase in the scale of collaborative robots affect the success rate of tasks?
4） The format of resumes is artificially defined. Will the format of resumes affect the efficiency of robot assistance? How to generate a better formatted resume?

---

> ### Author Response · Authors · 2024-11-21
>
> We really thank the reviewer for the time and efforts invested in this thoughtful evaluation of our paper. We would also appreciate the recognition of our work's strengths, particularly the acknowledgment of the innovative use of robot resumes generated from URDF files to enhance inter-robot communication and the effective leveraging of LLMs for heterogeneous multi-robot collaboration challenges. Apart from the positive comments, we would like to address the concerns and questions raised by the reviewer:
>
> ### **Comparison with similar frameworks and role-playing methods**
>
> We are really sorry that the writeup has led to some confusion. In the ablation study itself, we have already compared our framework with other similar baseline methods. For example,  **w/o. Robot resume** refers to **Meta GPT**[1] and **w/o. Discussion** refers to **CMAS**[2]. It is worth mentioning that Meta GPT[1] is a  role-playing method that assigns different roles (managers, developers, test engineers) manually to LLM agents as we discussed in the section 4.4 in the submission, lines [475 - 478]. Our method’s success rate, sub-goal success rate, and time efficiency all constantly outperform the role-playing method with human-created profiles. We really thank the reviewers for pointing out this ambiguity, and we will revise the draft as soon as possible.
>
> ### **Performance on various scenarios and different robot configurations**
>
> We greatly appreciate this question, though we feel that the scope of this question could be somewhat broad.
>
> First, we highlight that our Habitat-MAS benchmark is one of the most diverse and comprehensive datasets for involving 1) large-scale multi-floor multi-room indoor scenarios from Matterport3D [3] and HSSD [4]; 2) and diverse types of robots including drones, wheeled robot, legged robot, and different types of arms including revolute arms and prismatic arm, as we introduced in paper section 4.1, lines [372-379]. We believe our problem settings are representative and can cover a large ratio of indoor multi-robot scenarios and robot configurations that are **on the commercial market.**
>
> Yet we agree that this setting is still not the general scenarios nor general multi-robot configurations. However, due to the limited time for the rebuttal session and massive engineering efforts in designing new environments, tasks and robot teams, we cannot provide experimental results to answer this question. We also believe that this problem can be better addressed by large corporations and institutes with their resources and labor.
>
> ### **Question about extension to game relationship**
>
> Thank you for your insightful question regarding the potential extension of our framework to scenarios involving game relationships. While our current implementation focuses on a fully interconnected, collaborative robot system, we recognize the importance of addressing more complex scenarios.
>
> To extend our framework to game-theoretic situations, we propose two potential approaches:
>  - Introducing a group concept: We could incorporate the notion of groups or teams within the multi-robot system. This would allow us to model gaming relationships between different groups of interests, similar to scenarios found in competitive robotics applications like RoboMaster or robot soccer.
> - Implementing a more complex communication graph: Instead of a fully connected graph, we could design a layered communication topology. This approach would better represent scenarios where information flow is restricted or strategic, as often seen in game-theoretic contexts.
>
> However, we believe that the fundamental principles of our framework, particularly the embodiment-aware reasoning, could be adapted to handle these more complex interaction dynamics.
>
> [1] Sirui Hong, undefined., et al, "MetaGPT: Meta Programming for A Multi-Agent Collaborative Framework," in The Twelfth International Conference on Learning Representations, 2024.
>
> [2] Chen, Yongchao, Jacob Arkin, Yang Zhang, Nicholas Roy, and Chuchu Fan. "Scalable multi-robot collaboration with large language models: Centralized or decentralized systems?." In 2024 IEEE International Conference on Robotics and Automation (ICRA), pp. 4311-4317. IEEE, 2024.
>
> [3] Angel Chang, Angela Dai, Thomas Funkhouser, Maciej Halber, Matthias Niessner, Manolis Savva,
> Shuran Song, Andy Zeng, and Yinda Zhang. Matterport3d: Learning from rgb-d data in indoor
> environments. International Conference on 3D Vision (3DV), 2017
>
> [4] Khanna, Mukul, Yongsen Mao, Hanxiao Jiang, Sanjay Haresh, Brennan Shacklett, Dhruv Batra, Alexander Clegg, Eric Undersander, Angel X. Chang, and Manolis Savva. "Habitat synthetic scenes dataset (hssd-200): An analysis of 3d scene scale and realism tradeoffs for objectgoal navigation." In *Proceedings of the IEEE/CVF Conference on Computer Vision and Pattern Recognition.* 2024.

---

> > ### Author Response · Authors · 2024-11-21
> > **Official Comment by Authors (Part 2)**
> >
> > ### **Scalability of the framework**
> >
> > We recognize the importance of scalability of a multi-agent system and we are grateful to the reviewer for this question. Now we are emergently adding a new experiment to study the system's performance with the increase of robot numbers in the environment. We have started these two experiments:
> >
> > - In the first experiment, we are scaling the number of robots performing the same task. In our experiments, we sample 10 episodes from the manipulation task and evaluate performance across different numbers of Fetch robots (2, 4, 6, 10, and 20) to assess communication efficiency and success rate. However, due to computational limitations and the limited time of the rebuttal session, this experiment is still ongoing.
> > - In the second experiment, we plan to study the impact of increasing task complexity by scaling up the number of objects to manipulate. While maintaining a fixed number of Fetch robots (2), we evaluate the system's performance with varying numbers of objects (1, 2, 3, 5, and 10).
> >
> > Some existing results from the first ongoing experiment indicate that, with the key design of leader allocation and group discussion, the system can successfully assign a minimal yet sufficient number of robots to complete the manipulation tasks.
> >
> > We will report the experiment results with analysis in the revised paper and comment boxes as soon as possible.
> >
> > ### **Question on the impact of robot resume format on performance**
> > Thank you for your insightful question on the format of the robot resume. To answer the question, we conduct experiments among different formats of robot resumes (JSON, natural language, markdown, and XML), which are generated using GPT-4o:
> > - We sample 10 episodes from the perception task and evaluate the success rate of each format. The performance is mainly restricted by large language models' (GPT-4o) understanding capabilities.
> > - We present our experimental results on the success rate of each format setting below:
> > |  | JSON | Natural Language | Markdown | XML  |
> > | ------- | ---- | ---------------- | -------- | ---- |
> > | Average | 0.7  | 0.3              | 0.5      | 0.6  |
> > You can also download this [exp_robot_format.zip](https://github.com/EMOS-project/EMOS-project.github.io/blob/main/rebuttal/exp_robot_resume_format.zip) to check the data.
> > - According to our experiments, structured formats (JSON and XML) of robot resumes achieve higher success rates than unstructured formats (natural language). The success rates get higher as the format gets more structured.
> > - During experiments, we also found that some formats of robot resumes will lead to LLM hallucinations. In the setting of markdown and XML, the agents will fail to generate the correct format of actions, resulting in a 0% success rate. We then refine the prompt with minimum modification for meaningful results for comparison.
> >
> > To answer the question about how to generate a better-formatted resume, we suggest using structured formats like JSON format in our frameworks, rather than loosely structured formats like natural language.

---

> > > ### Comment · Reviewer_79K2 · 2024-11-22
> > >
> > > The author's response has resolved most of my confusion, but I would like to further discuss the last question with the author.
> > >
> > > The author mentioned that certain formats of resumes can cause LLM to be unable to understand. What are the drawbacks of these formats compared to other understandable formats? Can we conduct a simple analysis and find some commonalities, so that the LLM can understand all formats of resumes through certain techniques?

---

> > > > ### Author Response · Authors · 2024-11-23
> > > > **Reply to Question about Formats of Robot Resumes**
> > > >
> > > > We are very grateful for the reviewer’s feedback.
> > > >
> > > > As for the pros and cons of different format resumes for LLM reasoning, our experiments reveal that deeply nested XML or freeform natural language pose challenges for LLM comprehension due to issues like ambiguous field relationships or inconsistent representations, which could cause hallucination in embodied reasoning. In contrast, structured formats like JSON are more easily understood as they provide clear key-value mappings and reduce reliance on contextual reasoning.
> > > >
> > > > To address the common issues across formats, including lack of standardization and increased complexity in parsing, we could adopt a unified format specification to ensure consistent representation and use preprocessing techniques resulting in a JSON-like format to simplify complex format to a flat and structured form.

---

> > > > > ### Comment · Reviewer_79K2 · 2024-11-25
> > > > >
> > > > > Thanks for the author's response. I do not have any other issue for the submission, and this paper is suitable for ICLR.

---

> > > > > > ### Author Response · Authors · 2024-11-25
> > > > > >
> > > > > > We are really grateful to the reviewer's response and kind confirmation. And we truly appreciate the time and effort the reviewer has dedicated to thoroughly reviewing our work and engaging in in-depth discussions with us during rebuttal.

---

> > ### Comment · Reviewer_79K2 · 2024-11-22
> >
> > The author's response has resolved most of my confusion, but I would like to further discuss the last question with the author.
> >
> > The author is confident in the framework they have provided and proposes two possible methods to extend the proposed framework to game problems in incomplete communication situations. The reviewer would like to know what problems may be encountered when using these two methods. Can it still be simply transplanted over?

---

> > > ### Author Response · Authors · 2024-11-23
> > > **Reply to question about extension to game relationship**
> > >
> > > We really appreciate the in-depth discussion regarding the potential extension of our framework to scenarios involving game relationships among heterogeneous robots. Firstly, we are really sorry that we the authors are not experts in game theory or game relationships. Please feel free to correct us if we have inaccurate expressions. Here we provide our speculations about the potential challenges and problems in our proposed methods for adaptation:
> > >
> > > - Group-based interaction: Introducing groups may increase the complexity of agent policies. For example, agents should have different strategies for intra-group communication and inter-group communication depending on various game relationships and goals. Naively, we can transplant our framework to this scenario with different group system prompts. However, it is not clear to us how effective this naive transplantation could be.
> > > - Layered communication graph: A more complex communication graph also means higher computational complexity and a greater possibility of communication failure. The graph used in our current framework is a fully connected graph with a star topology. It will require some engineering efforts to transplant. Besides, sophisticated communication graph design and increase of agents in the network could lead to similar problems people facing in Computer Networks, which is beyond our knowledge boundary.

---

> ### Author Response · Authors · 2024-11-25
> **Update on Scalability Experiments' Results**
>
> We really thank the reviewer's insightful question on the scalability of the current framework. As promised before, we are now updating the results of the scalability experiments:
>
> ### **Table 1: Scalability with Increasing Agent Numbers**
>
> | Number of Robots | Success Rate (%) | Token Usage |
> |-------------------|------------------|-------------|
> | 2                | 80%             | 48779       |
> | 4                | 60%             | 73202       |
> | 6                | 70%             | 93252       |
> | 10               | 50%             | 151952      |
>
> In the first experiments of scaling up the robot number, we found that as the number scales up, the multi-agent system will face problems like hallucinations (in the setting of 10 agents) and the average success rate will decline. This is as expected since the hallucination problem in LLM is prevalent and it becomes worse with the increase of context length. This could be alleviated with more powerful LLM models as we have witnessed amazing progress in LLM model capability in the past year. Or designs like hierarchical communication with smaller sub-group discussions and larger group aggregation (delegate meeting) could help to solve the scalability problem in multi-agent discussion.
>
>
> ### **Table 2: Scalability with Increasing Task Complexity**
>
> | Number of Objects | Success Rate (%) | Token Usage |
> |--------------------|------------------|-------------|
> | 1                 | 90%             | 25778       |
> | 2                 | 80%             | 50005       |
> | 3                 | 80%             | 87668       |
> | 5                 | 70%             | 197485      |
>
> In the second experiment, which involved scaling up task complexity, we found that our system demonstrates robustness to a certain extent. As the number of objects increases—representing greater task complexity—the system maintains a relatively high and stable success rate (above 70%). This indicates that our communication structure is effective and scalable, capable of handling more complex problems.
>
> For the experiments raw data, you can check this file: [exp_scalability.zip](https://github.com/EMOS-project/EMOS-project.github.io/blob/main/rebuttal/exp_scalability.zip)
>
> ### **Explanation of changing experiment setting**
> We are conducting this experiment on a workstation with a single RTX 4090. Due to the limitations of the simulator and computational constraints, rendering camera sensors for 20 robots always caused the program to crash. If time permits, there might be a workaround for the simulation to reduce the resource consumption. As a result, we simplified the experimental settings and removed the setting for 20 robots.
>
> We thanks again for your thoughtful question.

---

> > ### Comment · Reviewer_79K2 · 2024-11-26
> >
> > Thanks for the author's effort. The additional experiments solve my confusion. I do not have any problem.

---

### Official Review · Reviewer_5WcA · 2024-11-02

**Soundness:** 4
**Presentation:** 4
**Contribution:** 3
**Rating:** 6
**Confidence:** 4

**Summary:**

This paper presents the Embodiment-Aware Heterogeneous Multi-Robot Operating System (EMOS), an LLM-driven, multi-agent system designed to manage diverse robots in complex household tasks.

**Strengths:**

The paper is well-written and explores an interesting application.  In essence, it describes a system with multiple robots, each equipped with an LLM and distinct capabilities. The robots communicate to determine task distribution, using robot resumes to identify which robot is best suited for each task. Through hierarchical and decentralized planning, they collaborate to complete complex tasks in a shared environment, leveraging spatial reasoning and embodiment awareness to enhance coordination.
1) The main contribution appears to center around how questions are passed to the pre-trained LLM for task allocation in a multi-robot system.  Authors proposed a robot resume method, that creates a dynamic capability profile for each robot. This profile includes an understanding of each robot's URDF files, enabling the system to call upon robot kinematics tools to generate detailed descriptions of their physics capabilities.
2) Habitat-MAS simulation: It is designed for multi-agent systems, facilitating complex task management using heterogeneous team of robots.

**Weaknesses:**

Presenting results is not a contribution. Instead, it serves as a validation that the proposed methodology works as intended and is better than baselines.

**Questions:**

I don't have any questions.

---

> ### Author Response · Authors · 2024-11-20
> **Rebuttal comment**
>
> We thank the reviewer for the time and effort you have invested in evaluating our work. We are grateful for the positive comments on the soundness and presentation of our paper. Although the reviewer thinks differently and **“would recommend the authors to submit it for a Robotics conference”**, we would like to address the reviewers’ concerns and do more clarifications:
>
> ### **Relevance to ICLR scope**
>
> Firstly, we would like to clarify why this paper is relevant to ICLR scope, in response to the reviewer's comment to submit to the Robotics conference. Due to the length limitation of the comment box, we have written a comprehensive relevance statement, available at [Relevance Statement](https://github.com/EMOS-project/EMOS-project.github.io/blob/main/relevance_statement/relevance_statement.pdf). You can also check all the open-source data and data processing script in https://github.com/EMOS-project/EMOS-project.github.io/tree/main/relevance_statement
>
> In short, our work aligns closely with ICLR's growing focus on robotics, embodied AI, and benchmarking, as evidenced by the increasing number of accepted papers in these areas over the past few years.
>
> ### [(CLICK) statistics plot](https://github.com/EMOS-project/EMOS-project.github.io/blob/main/relevance_statement/paper_statistics.png)
>
> We highlight that, only in ICLR 2024, there are 83 robotics and 103 benchmark papers ACCEPTED. EMOS addresses key challenges in embodiment-aware task planning that are currently missing in the Embodied AI community, making it highly relevant to ICLR's scope.
>
> ### **Regarding the novelty of our contribution**
>
> While we understand the reviewer’ s perspective, we believe our work presents significant innovations in the field of embodied AI and multi-robot systems. EMOS addresses a crucial gap in embodiment-aware task planning for heterogeneous multi-robot systems, which has not been adequately explored in previous research. Our approach of leveraging large language models (LLMs) for this purpose is novel and aligns with the growing focus on robotics and embodied AI at ICLR, as evidenced by recent accepted papers in this domain.
>
> ### **Suggestion of fine-tuning the LLM**
>
> We appreciate your suggestion to explore custom fine-tuning of the LLM to be tailored for our application. However, the focus of our paper is leveraging the strong priors and reasoning capabilities of pre-trained LLMs and design of LLM multi-agent systems to solve way too complex problems for a single LLM model, i.e. in the context of “LLM agents”. We agree that further adaptation could enhance performance, and we plan to explore this direction in future work.
>
> ### **Contribution of Habitat-MAS simulation**
>
> The purpose of this benchmark platform extends beyond mere implementation. It is for the broader Embodied AI community, enabling researchers to study and develop methods for further automation of complex multi-robot systems in a more general problem setting. **As we stated in lines [92-96] of the submission paper, our benchmark tasks are processed such that not all robots or random agents can finish a specific task.** The LLM agents need to understand their physical capabilities for the task planning.
>
> Finally, we really thank the reviewer's time and effort in reviewing this paper. We also value the different perspectives the reviewer held at the beginning. We do hope the statistics and clarifications above can help us win the reviewer back. We would really appreciate it if the reviewer is willing to have more discussions here and we will respond with our best effort.

---

> > ### Author Response · Authors · 2024-11-21
> > **Relevance Statement**
> >
> > For the convenience of the reviewer and AC, we have pasted the content of the Relevance Statement under this comment for reference.
> >
> > ### **Relevance Statement**
> >
> > We would like to state that our work on the Embodiment-Aware Heterogeneous Multi-Robot Operating System (EMOS) aligns closely with the scope of ICLR, particularly in areas like robotics, embodied AI, and benchmarking.
> >
> > According to statistics about **ICLR accepted papers from 2022 to 2024** (data source is from [papercopilot](https://papercopilot.com/statistics/iclr-statistics/), there has been a growing focus on these topics, which can referred to the [figure 1](https://github.com/EMOS-project/EMOS-project.github.io/blob/main/relevance_statement/paper_statistics.png?raw=true). This figure shows the filters relevant past ICLR papers and visualizes trends in topics such as Robotics and Benchmarks, verifying the alignment of our work with the increasing focus of ICLR community.
> >
> > Then, we will specifically list and analyze some of the works in the past few years at ICLR that are similar to the topic or contributions of our article.Large-scale pretrained models, especially LLMs, trained on internet-scale datasets, bring strong priors and reasoning capabilities to everyday tasks. Many interesting ICLR papers have explored LLM applications for robotic manipulation and other embodied agent tasks. Notable among ICLR 2024:
> >
> > - "**Programmatically Grounded, Compositionally Generalizable Robotic Manipulation**", which leverages pre-trained model modularity to aid robotic manipulation.
> > - "**Building Cooperative Embodied Agents Modularly with Large Language Models**", which uses LLMs as different modules within multi-agent systems.
> > - "**Steve-Eye: Equipping LLM-based Embodied Agents with Visual Perception in Open Worlds**", an end-to-end LLM-based model enabling embodied agents to perceive their environment.
> > - "**Habitat 3.0: A Co-Habitat for Humans, Avatars, and Robots**", presents a simulation platform for human-robot interaction tasks.
> >
> > Additionally,
> > - "**GenSim: Generating Robotic Simulation Tasks via Large Language Models**", applies LLMs to robotic task generation and policy learning.
> > - "**Vision-Language Foundation Models as Effective Robot Imitators**", fine-tunes large models for robot gripper control, both highly judged as spotlight papers by the ACs.
> >
> > In the area of agent benchmarking and multi-agent systems, exciting works include:
> > - "**LoTa-Bench: Benchmarking Language-oriented Task Planners for Embodied Agents**"
> > - and "**SmartPlay: A Benchmark for LLMs as Intelligent Agents**", all explore various aspects of LLM evaluation.
> > - Of particular note, the oral paper "**MetaGPT: Meta Programming for A Multi-Agent Collaborative Framework**” introduces an innovative role-play approach for multi-agent collaboration.
> >
> > Our work aims to identify the problem of embodiment-aware task planning that is still missing in the Embodeid-AI community, providing a platform to study this problem and provide an initial method to try to address these challenges in heterogeneous multi-robot systems. While it’s true that a large proportion of our efforts fall in simulation engineering for the benchmark, one of the main motivations is to provide a platform to share with the Embodied AI community to work on this direction of further automation of complex multi-robot systems with a more general problem setting. In the methodology part, by leveraging the priors that LMs hold about robot capabilities parsing from URDF and their decision-making capabilities in complex tasks, we address a practical issue of embodiment-aware understanding innovatively. By building a hierarchical multi-agent system that leverages the synchronized communication and distributed execution, we provide a practical approach that is tailored for real-time multi-robot system settings, and this could serve as a starting point for more comprehensive system development.
> >
> > In conclusion, our work on EMOS aligns with ICLR's growing focus on robotics and embodied AI, addressing the crucial gap in embodiment-aware task planning for heterogeneous multi-robot systems, while providing a benchmark platform and innovative methodology leveraging large language models.

---

> > > ### Comment · Reviewer_5WcA · 2024-11-24
> > > **Thanks for the relevance statement**
> > >
> > > Dear Authors,
> > >
> > > Thanks for the detailed relevance statement, upon checking the statistics I am convinced that this paper is indeed suitable for ICLR. I will change my prior opinion and edit the review accordingly. Aside from the initial concerns about relevance, I find no other issues with the submission.

---

> > > > ### Author Response · Authors · 2024-11-25
> > > >
> > > > We really thank the reviewer's time and effort in reviewing our materials to clarify this issue. We are also happy to see the decision by the reviewer to accept our paper.

---

> ### Author Response · Authors · 2024-11-23
> **Kind reminder for review response**
>
> Dear reviewer,
>
> We hope this message finds you well. We are writing to request your response to our rebuttal comment kindly. We would appreciate it if you could review our response and we could have helpful discussions here.
>
> Kind regards,
>
> The authors

---

### Author Response · Authors · 2024-11-23
**Update of Revision V1**

Dear reviewers,

We greatly appreciate all the constructive comments from the reviewers. We have revised our paper according to the suggestions. The updates are as follows:
- We reorganized the `Section 2. Related Works` and their relations to our paper as suggested by **Reviewer jjgY** and **Reviewer xZUU**. Specifically, we divided the "Multi-Agent System" subsection into two sections "LLM-Based Multi-Agent System" and "Heterogeneous Multi-Agent Learning". we also updated the "multi-robot systems" section with new references. All the mentioned related works by these two reviewers have been added to the reference list.
- We added `Appendix A.2 MULTI-AGENT SYSTEM DESIGN AND COMMUNICATION` according to the request by **Reviewer xZUU** to add the agents' communication graph
- We added `Appendix B.4 DETAILED IMPLEMENTATION OF ROBOT LOW-LEVEL CONTROL IN BENCHMARK` according to the request by **Reviewer xZUU**.
- We added `Appendix C.5 EXTRA EXPERIMENT ON FORMAT OF ROBOT RESUME` according to the question by **Reviewer 79K2**

We have highlighted all the revised/added parts with blue color for the reviewers' convenience. We will remove the colorization in the formal version. We are also looking forward to reviewers' feedback on this revision.

---

> ### Author Response · Authors · 2024-11-27
> **Update of Revision V2**
>
> Dear reviewers,
>
> We greatly appreciate all the constructive comments from the reviewers. We have revised our paper according to the suggestions. Based on the revision V1, we have these extra updates:
> - We added `C.6 EXTRA EXPERIMENTS ON SCALABILITY OF EMOS`. Thanks to insightful questions by `Reviewer 79K2` and `Reviewer jjgY` about the scalability of the EMOS framework, we had helpful discussions about this issue. In this revision, we summarized the extra experiments and discussions about scalability to a new section in appendix to discuss this issue.
>
>
> We have highlighted all the revised/added parts with blue color for the reviewers' convenience. We will remove the colorization in the formal version. We are also looking forward to reviewers' feedback on this revision.

---

### Meta-Review · Area_Chair_v3Tx · 2024-12-17

**Metareview:**

The paper proposes EMOS, a framework that uses large language models (LLMs) to dynamically coordinate the behavior of teams of heterogeneous robots according to their physical design. EMOS generates a "robot resume" based upon a robot's kinematics (as determined by its URDF) that it then uses to determine their roles. Additionally, the paper proposes a new benchmark (Habitat-MAS) for evaluating multi-agent coordination for tasks that involve manipulation, navigation, and object rearrangement within a multi-floor building.

The paper was reviewed by four referees who largely agreed on the paper's strengths an weaknesses. Among the strengths, several reviewers appreciated the use of LLMs to dynamically allocate tasks across robot teams based upon their kinematics, as opposed to the more traditional approach of pre-defining their roles. At the same time, several reviewers shared the concern that the paper read more like a project report that described a particular prompting strategy, as opposed to a research paper, and that it lacked several key details about the framework. In their response to the reviewers, the authors addressed the need for more detailed explanations of EMOS, however they did not speak to the concern about the presentation style. Meanwhile, at least two reviewers emphasized the need to compare against traditional role playing approaches, while one asked for an evaluation of how the method scales as the number of robots increases and the tasks become more complex. The authors clarified that the initial submission already provided a comparison to role playing approaches, while they provided an additional analysis of scalability during the rebuttal period.

**Additional Comments On Reviewer Discussion:**

There was a healthy amount of discussion between the authors, reviewers, and the AC. This included a discussion of concerns with can initial review that questioned the relevance of a systems-focused paper for ICLR as well as statements that the authors found to be lack adequate justification. The AC discussed these concerns with the reviewer, who then updated their review.

---

### Decision · Program_Chairs · 2025-01-22

Accept (Poster)